# Proteasomal degradation induced by DPP9-mediated processing competes with mitochondrial protein import

Yannik Finger[1,†], Markus Habich[1,†], Sarah Gerlich[1], Sophia Urbanczyk[2], Erik van de Logt[1], Julian Koch[1] ORCID, Laura Schu[1], Kim Jasmin Lapacz[1], Muna Ali[1], Carmelina Petrungaro[1], Silja Lucia Salscheider[1], Christian Pichlo[3], Ulrich Baumann[3], Dirk Mielenz[2], Joern Dengjel[4] ORCID, Bent Brachvogel[5,6], Kay Hofmann[7] ORCID & Jan Riemer[1,8,*] ORCID

## Abstract

**Plasticity of the proteome is critical to adapt to varying conditions. Control of mitochondrial protein import contributes to this plasticity. Here, we identified a pathway that regulates mitochondrial protein import by regulated N-terminal processing. We demonstrate that dipeptidyl peptidases 8/9 (DPP8/9) mediate the N-terminal processing of adenylate kinase 2 (AK2) en route to mitochondria. We show that AK2 is a substrate of the mitochondrial disulfide relay, thus lacking an N-terminal mitochondrial targeting sequence and undergoing comparatively slow import. DPP9-mediated processing of AK2 induces its rapid proteasomal degradation and prevents cytosolic accumulation of enzymatically active AK2. Besides AK2, we identify more than 100 mitochondrial proteins with putative DPP8/9 recognition sites and demonstrate that DPP8/9 influence the cellular levels of a number of these proteins. Collectively, we provide in this study a conceptual framework on how regulated cytosolic processing controls levels of mitochondrial proteins as well as their dual localization to mitochondria and other compartments.**

**Keywords** adenylate kinase 2; dipeptidyl peptidase 9; MIA40; mitochondrial protein import; quality control
**Subject Category** Membrane & Trafficking
**The EMBO Journal (2020) 39: e103889**

## Introduction

Mitochondria fulfill crucial functions in metabolism and signaling. They contain more than 1,500 proteins of which only a small number is synthesized in the organelle (Rhee *et al*, 2013; Hung *et al*, 2014; Calvo *et al*, 2016; Morgenstern *et al*, 2017). The remainder is translated on cytosolic ribosomes and relies on different mitochondrial import pathways to reach the mitochondrial subcompartments, namely the mitochondrial outer (OMM) and inner (IMM) membranes, the intermembrane space (IMS) and the mitochondrial matrix (Topf *et al*, 2016; MacPherson & Tokatlidis, 2017; Jackson *et al*, 2018; Habich *et al*, 2019b; Hansen & Herrmann, 2019; Pfanner *et al*, 2019).

Mitochondrial biogenesis and other cellular processes are tightly coordinated to adjust cells to different metabolic needs (Harbauer *et al*, 2014; Chandel, 2015; Finkel, 2015; Wai & Langer, 2016). Coordination thereby takes place on different levels including regulation of transcription and translation of mitochondrial proteins. Posttranslational regulation often directly targets mitochondrial precursor proteins in the cytosol and modulates their mitochondrial import. This includes the binding of metabolites or partner proteins to these precursors (Vongsamphanh *et al*, 2001; Dailey *et al*, 2005; Frechin *et al*, 2009) and the attachment of covalent modifications or the proteolytic processing of precursor proteins (Boopathi *et al*, 2008; DeRasmo *et al*, 2008; Avadhani *et al*, 2011; Bragoszewski *et al*, 2013). Moreover, competition between cytosolic folding and mitochondrial import has been reported as regulatory principle for a number of dually localized proteins (Strobel *et al*, 2002; Kloppel *et al*, 2011; Yogev & Pines, 2011; Suzuki *et al*, 2013). Folding in the cytosol prevents mitochondrial import, and thus, the majority of the

1  Institute of Biochemistry, Redox Biochemistry, University of Cologne, Cologne, Germany
2  Division of Molecular Immunology, Department of Internal Medicine III, Nikolaus-Fiebiger-Center, University of Erlangen-Nürnberg, Erlangen, Germany
3  Institute of Biochemistry, University of Cologne, Cologne, Germany
4  Department of Biology, University of Fribourg, Fribourg, Switzerland
5  Department of Pediatrics and Adolescent Medicine, Experimental Neonatology, Faculty of Medicine, University of Cologne, Cologne, Germany
6  Center for Biochemistry, Faculty of Medicine, University of Cologne, Cologne, Germany
7  Institute of Genetics, University of Cologne, Cologne, Germany
8  Cologne Excellence Cluster on Cellular Stress Responses in Aging-Associated Diseases (CECAD), University of Cologne, Cologne, Germany
   *Corresponding author. Tel: +49 221 470 7306; E-mail: jan.riemer@uni-koeln.de
   †These authors contributed equally to this work

protein ends up localizing in the cytosol and not in mitochondria. Posttranslational modification of the translocase of the OMM (TOM) that constitutes the entry gate for the vast majority of mitochondria precursors can as well influence import of specific proteins (Schmidt *et al*, 2011; Gerbeth *et al*, 2013).

Precursor proteins are handed over to different import machineries after passage of the TOM pore. One such machinery is the mitochondrial disulfide relay that imports and oxidatively folds proteins of the IMS (Stojanovski *et al*, 2012; MacPherson & Tokatlidis, 2017; Habich *et al*, 2019b). The protein MIA40 (also coiled-coil-helix-coiled-coil-helix domain containing protein 4, CHCHD4) is the main component of the mitochondrial disulfide relay and serves as oxidoreductase and import receptor (Petrungaro *et al*, 2015; Peleh *et al*, 2016; Habich *et al*, 2019a). Most substrates of the disulfide relay lack classical N-terminal mitochondrial targeting sequences (MTS) and instead contain conserved cysteine motifs (Fischer *et al*, 2013; Petrungaro *et al*, 2015; Modjtahedi *et al*, 2016; Bragoszewski *et al*, 2017; Hansen & Herrmann, 2019). The majority of substrates contains four conserved cysteines in so-called twin $CX_9C$ or twin $CX_3C$ motifs that become oxidized to two very stable disulfide bonds during substrate maturation by MIA40. Recently, few unconventional substrates of the disulfide relay with differing structures and cysteine patterns have been identified (Barchiesi *et al*, 2015; Petrungaro *et al*, 2015; Ramesh *et al*, 2016). One such putative substrate that occurred as interaction partner of MIA40 but has so far not been characterized was adenylate kinase 2 (AK2). AK2 catalyzes the reversible adenine nucleotide phosphoryl transfer in the reaction 2 ADP ↔ ATP + AMP in the IMS. Loss of AK2 leads to impaired mitochondrial function (Burkart *et al*, 2011; Six *et al*, 2015), hampers induction of the endoplasmic reticulum unfolded protein response (Burkart *et al*, 2011), and sensitizes cells to induction of apoptosis (Single *et al*, 1998; Kohler *et al*, 1999; Lee *et al*, 2007). Human patients with defects in AK2 suffer from an autosomal recessive form of severe combined immunodeficiency (SCID) named reticular dysgenesis (RD) (Lagresle-Peyrou *et al*, 2009; Pannicke *et al*, 2009). Although changes in AK2 activity can have severe consequences, the mechanism of AK2 IMS import and folding remains unclear.

In this study, we show that mitochondrial import of AK2 is mediated by the mitochondrial disulfide relay via an unusual intramolecular disulfide shuffling mechanism that makes use of the three conserved cysteines in AK2. We also demonstrate that N-terminal processing of AK2 by the dipeptidyl peptidases DPP8 and DPP9 controls its IMS import and cellular localization. Processed cytosolic AK2 is thereby targeted for rapid proteasomal degradation. Attenuation of processing results in increased amounts and activity of AK2 and induces dual localization to IMS and cytosol of previously exclusively IMS-localized AK2. Lastly, we show that processing by DPP8 and DPP9 is employed as general control mechanism of mitochondrial protein biogenesis and might affect more than 100 mitochondrial proteins to regulate mitochondrial proteome plasticity.

## Results

### AK2 is oxidized by MIA40 during its maturation

AK2 is a soluble IMS protein (Bruns & Regina, 1977; Kohler *et al*, 1999). It contains conserved cysteines (C40, C42, and C92) and

lacks an N-terminal MTS (Fig 1A). Two AK2 isoforms (AK2-A and AK2-B) can be detected in HEK293 cells. These isoforms differ in the most C-terminal amino acids and thereby AK2-A contains one additional non-conserved cysteine (C232) (Fig 1A and Appendix Fig S1A). We identified AK2 as an interaction partner of MIA40 in proteomics screens (Petrungaro *et al*, 2015; Habich *et al*, 2019a). To confirm this interaction, we precipitated C-terminally Strep-tagged MIA40 variants from HEK293 cells and tested whether endogenous AK2 co-precipitated by immunoblot. We confirmed the AK2-MIA40 interaction but also observed that the interaction depended on the redox-active cysteines of MIA40 as no AK2 was co-precipitated with a MIA40 lacking these cysteines (Fig 1B). We also confirmed the interaction by performing the inverse experiment, precipitating C-terminally HA-tagged AK2 and testing for coprecipitation of endogenous MIA40 (Appendix Fig S1B), and by precipitating endogenous MIA40 and testing for coprecipitation of endogenous AK2 (Appendix Fig S1C). Lastly, we demonstrated by emetine chase and pulse-chase analyses that the AK2-MIA40 interaction was transient in nature (Appendix Fig S1D and E). This indicated that AK2 might undergo MIA40-catalyzed oxidative protein folding during its maturation. To test this, we assessed different parameters for such a maturation pathway.

First, we determined the redox state of the cysteines of mature endogenous AK2 using an inverse redox state assay (Erdogan *et al*, 2018) (Fig 1C and Appendix Fig S1F). In this assay, the thiol–disulfide redox state of proteins is "trapped" in intact cells by the addition of the membrane-permeable alkylating agent N-ethyl maleimide (NEM). This modifies all accessible (reduced) thiols. Thereafter, cells are lysed and previously oxidized cysteines are reduced using the reductant tris(2-carboxyethyl)phosphine (TCEP). Then, free thiols (the ones that had been previously oxidized) are modified using methyl-PEG-12-maleimide (mmPEG12). This compound introduces a molecular weight shift that can be detected on SDS–PAGE. Thus, proteins that contained oxidized cysteines in intact cells will migrate slower on SDS–PAGE compared to previously reduced proteins. When we applied this assay to HEK293 cells and analyzed the redox state of AK2, we found that both isoforms, AK2-A and AK2-B, migrated slower than the purely NEM-treated control indicating that they contained oxidized cysteines, most likely in one disulfide bond (Fig 1C, lanes 2 and 3, black arrow heads, and Appendix Fig S1F).

Next, we investigated whether the disulfide bond in AK2 formed during AK2 maturation (Fig 1D). To this end, we employed an *in vivo* oxidation assay (Fischer *et al*, 2013). In this assay, we combined a pulse-chase experiment with a redox state determination and immunoprecipitation (IP) of the protein of interest. Pulse labeling with [35]S-methionine thereby allowed the labeling of a small fraction of newly synthesized proteins. In the subsequent chase period, the fate of this fraction can be followed. We did so by freezing the thiol–disulfide redox state by rapid acidification using trichloroacetic acid (TCA). Then, free thiols were modified with mmPEG12 before the protein of interest; in our case AK2, was immunoprecipitated. Subsequently, modified proteins were analyzed by SDS–PAGE and autoradiography. Reduced proteins thereby migrate slower than oxidized ones. When we employed this assay to AK2, we observed that both isoforms of endogenous AK2 migrate faster with increasing chase time. This indicates that they acquire an intramolecular disulfide bond with time (Fig 1D).

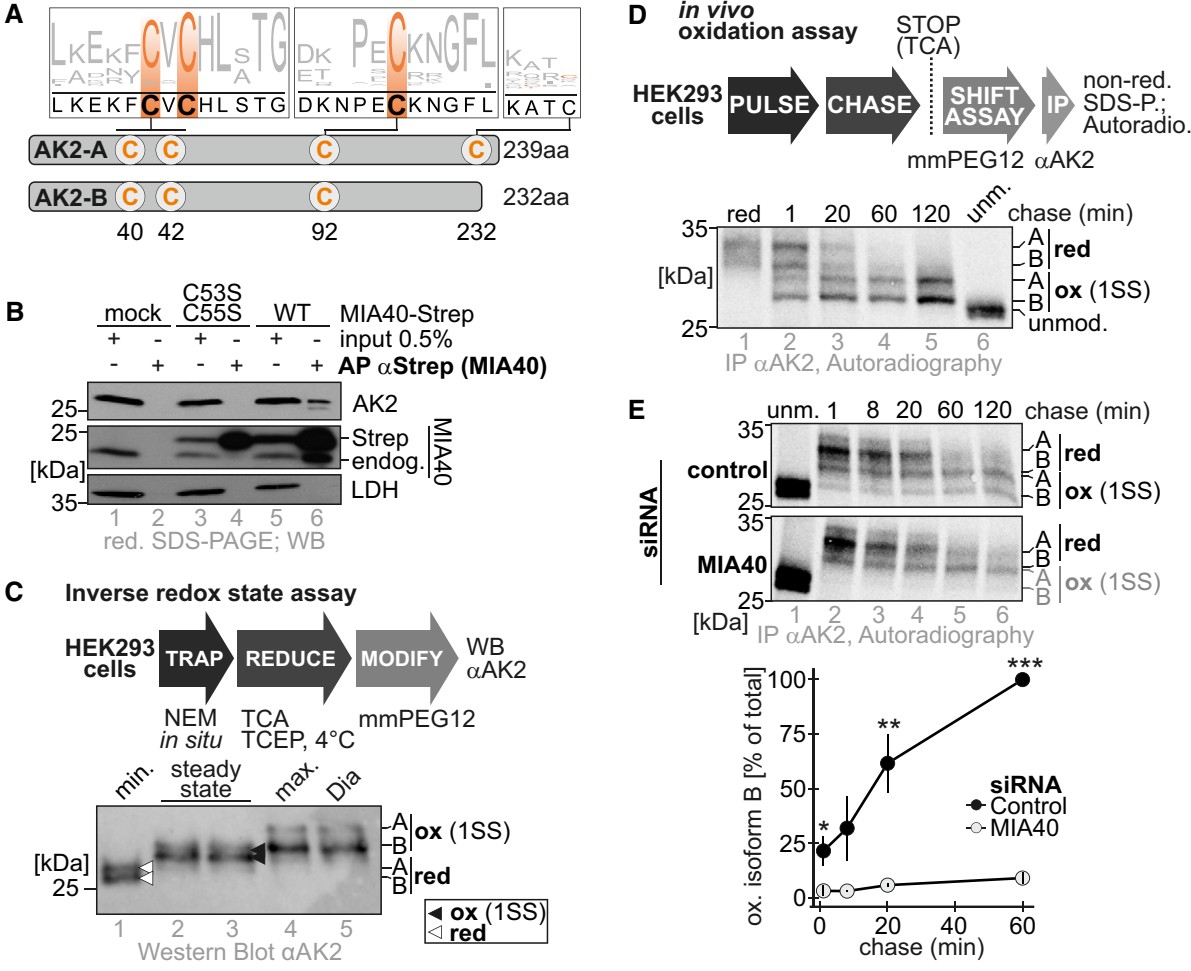

**Figure 1. AK2 is a substrate of the mitochondrial disulfide relay.**

A  Cysteine conservation of AK2. AK2 contains four cysteines of which three (C40, C42, and C92) are highly conserved. AK2 exists in two isoforms that differ in the C-terminus with the "A" isoform containing C232 and the "B" isoform not. AK2 does not contain a mitochondrial targeting sequence. Logo plot was derived from alignments of AK2 from 163 species.

B  Interaction analysis for MIA40 and AK2. Stable inducible cell lines that express either different variants of MIA40-Strep (wild type and a redox inactive mutant MIA40$^{C53S,C55S}$), (Fischer *et al*, 2013) or AK2-HA were treated with N-ethyl maleimide (NEM) to stop thiol–disulfide exchange processes and lysed under denaturing conditions. Lysates were analyzed by subsequent affinity purification against Strep (MIA40) and immunoblotting. Like classical MIA40 substrates, AK2 interacted with the active site of MIA40 (lane 6). 0.5% input was loaded as control.

C  Redox state analysis of endogenous AK2 by inverse redox shift assay. HEK293 cells were lysed by TCA precipitation after NEM treatment, and lysates were reduced using TCEP and then thiols modified using the alkylating agent mmPEG$_{12}$. With this "inverse" method, previously oxidized cysteine thiols are modified with mmPEG$_{12}$, which results in a slower migration on SDS–PAGE. Treatment of intact cells with diamide and consequently modification of all cysteines served as control. Likewise, absence of NEM treatment served as "max" control. Treatment lysates with NEM instead of mmPEG$_{12}$ served as minimal shift (min) control. A, B, isoforms of AK2. Both isoforms of AK2 are semi-oxidized in intact cells.

D  Oxidation kinetics experiment to investigate maturation of endogenous AK2. HEK293 cells were pulse-labeled for 10 min with [$^{35}$S]-methionine and chased with cold methionine for the indicated times. The chase was stopped by TCA precipitation, and then, the lysate was treated with mmPEG$_{12}$ to determine the AK2 redox states, followed by IP against AK2. Eluates were analyzed by Tris–Tricine–PAGE and autoradiography. With this "direct" redox state determination method, reduced AK2 was modified with either three (isoform B) or four (isoform A) mmPEG$_{12}$. Semi-oxidized AK2 was modified with only one ("B") or two ("A") mmPEG$_{12}$, respectively. Unmodified and fully reduced cell lysates served as controls. Endogenous AK2 becomes oxidized with a half time of less than 20 min.

E  Oxidation kinetics experiment to investigate the influence of MIA40 on AK2 maturation. Experiment was performed as described in (D), except that 72 h before experiments cells were transfected with control siRNA or siRNA directed against MIA40. Endogenous AK2 oxidation depends on MIA40. Reported values are the mean of 3 independent experiments; error bars represent ± SD. Student's *t*-test was performed. * represents $P < 0.05$, ** represents $P < 0.01$, and *** represents $P < 0.001$.

Source data are available online for this figure.

Finally, we assessed whether depletion of MIA40 by siRNA-mediated knockdown would interfere with AK2 oxidation in intact cells (Fig 1E and Appendix Fig S1G). We repeated the *in vivo* oxidation assay in cells depleted for MIA40. While cells treated with control siRNA exhibited normal oxidation kinetics of endogenous AK2, cells treated with siRNA directed against MIA40 failed to oxidize AK2 (Fig 1E). Taken together, we conclude that MIA40 oxidizes AK2 and thereby introduces one disulfide bond into the protein.

## Disulfide bond formation between C42 and C92 is critical for mitochondrial AK2 accumulation

Many classical MIA40 substrates contain an even number of conserved cysteines in a specific defined spacing. AK2 seems to be an unconventional substrate of MIA40 as it is not only comparatively large (*ca.* 26 kDa) but has an odd number of conserved cysteines (C40, C42, and C92) with an unconventional spacing. Of these cysteines, only C40 and C92 are positioned in helices and thus constitute classical targets for initial MIA40 interaction (Milenkovic *et al*, 2009; Sideris *et al*, 2009; Banci *et al*, 2010; Koch & Schmid, 2014). We assessed the role of AK2 cysteines for oxidation and mitochondrial accumulation of the protein.

First, we tested the thiol–disulfide redox state of the different AK2 cysteine variants (Fig 2A). To this end, we generated stable inducible cell lines, which express different C-terminally HA-tagged AK2 isoform A variants (AK2$^{WT}$, AK2$^{C40S}$, AK2$^{C42S}$, AK2$^{C92S}$, AK2$^{C232S}$). Compared to the redox state determination experiment with endogenous AK2, we therefore only detect the isoform AK2-A. Wild-type AK2-HA and AK2$^{C232S}$-HA were present in an overwhelmingly oxidized redox state. AK2$^{C40S}$-HA was mainly oxidized but exhibited a considerable reduced fraction at steady state. Conversely, AK2$^{C42S}$-HA and AK2$^{C92S}$-HA were present in the completely reduced state indicating that in the wild-type protein a disulfide bond is present between C42 and C92 (Fig 2A). This is also in agreement with the AK2 crystal structure (Fig 2B, pdb: 2C9Y). The electron density and the deposited coordinates of this entry indicated a partial disulfide bond between C42 and C92 with an occupancy of about 0.5. The imperfect formation of the disulfide bond in the crystal structure is probably due to the fact that during the preparation of recombinant protein for crystallization no care was taken to preserve a native disulfide pattern and possibly a partial photoreduction during X-ray exposure may have taken place. Nevertheless, the structure together with our data provides strong evidence for the presence of a disulfide bond connecting C42 and C92.

Cysteine oxidation and mitochondrial accumulation are closely coupled events for MIA40 substrates without MTS (Habich *et al*, 2019a). Therefore, a lack of cysteines does not only result in the absence of oxidation but also in cytosolic mislocalization (Friederich *et al*, 2017; Mohanraj *et al*, 2019). We thus next investigated mitochondrial localization of AK2 cysteine variants by immunofluorescence microscopy (Fig 2C, and Appendix Fig S2A and B). Wild-type AK2-HA and AK2$^{C232S}$-HA colocalized with a mitochondrial marker. Conversely, AK2$^{C42S}$-HA and AK2$^{C92S}$-HA did not colocalize with mitochondria and instead appeared to be cytosolic. This indicated that these two cysteines and the formation of the disulfide bond are critical for mitochondrial accumulation. Interestingly, AK2$^{C40S}$-HA exhibited an intermediate behavior. It was colocalized with mitochondria to a significantly lower extent compared to wild-type AK2. Still the majority of the protein appeared to localize to mitochondria (Fig 2C).

Collectively, these data indicate that formation of a disulfide bond between C42 and C92 is critical for mitochondrial accumulation of AK2. C40 appears to exert an influence on mitochondrial accumulation and AK2 redox state as well.

## C40 drives intramolecular disulfide shuffling dependent import of AK2

Intrigued by the behavior of AK2$^{C40S}$-HA, we investigated this variant in more detail. In the AK2 structure, C40 is uniquely positioned in a motif that is suitable for interaction with MIA40 (Sideris *et al*, 2009; Koch & Schmid, 2014) (Appendix Fig S2C). This motif consists of an amphipathic helix with hydrophobic patches on the helix face that is marked by the presence of the MIA40-interacting cysteine. Although C92 is also positioned in a helix, this helix lacks the amphipathic character required for MIA40 interaction. Thus, the two cysteines (C42 and C92) that eventually form a disulfide bond in mature AK2 are not positioned in an environment that is predicted to facilitate efficient interaction with MIA40. Does this point to a role of C40 in initiating interaction between AK2 and MIA40? To answer this question, we immunoprecipitated endogenous MIA40 from cell lines expressing either AK2$^{WT}$-HA or AK2$^{C40S}$-HA, and analyzed the eluates by immunoblot against the HA tag. We thereby found that absence of C40 strongly interfered with the AK2-MIA40 interaction (Appendix Fig S2D). If C40 is so important for initiating the MIA40-AK2 interaction, its absence should also affect oxidative folding (i.e., maturation) of AK2. Indeed, when we analyzed the oxidation kinetics of AK2 variants, we found that AK2$^{C40S}$ has a delayed oxidation compared to wild-type AK2 (Appendix Fig S2E).

Our findings point to an initial interaction between C40 and MIA40. This mixed disulfide bond would have to isomerize to the final disulfide bond between C42 and C92. We hypothesized that this might take place via an intramolecular disulfide isomerization mechanism. Although such a process has never been described for mitochondrial disulfide relay substrates, it is well known for oxidative folding of complex proteins in the endoplasmic reticulum (ER) (Jansens *et al*, 2002; Land *et al*, 2003; Chakravarthi & Bulleid, 2004; Roberts *et al*, 2018). The temporal resolution of our *in vivo* pulse-chase experiments was not sufficient to address the question of intramolecular disulfide shuffling in AK2. We thus turned to an *in vitro* experiment (Appendix Fig S2F). With this experiment, we tested whether in principle disulfide formation between C40 and, e.g., C42 is possible. This disulfide bond would be a critical intermediate in such an isomerization process. We purified different cysteine variants of AK2 and analyzed their disulfide pattern as well as the stability of the disulfide bonds (Appendix Fig S2F). We indeed found that all tested variants except for the AK2$^{C40S,C42S}$ contained a disulfide bond. This also included the AK2$^{C92S,C232S}$ variant that can only form a disulfide bond between C40 and C42. In line with this, also the C92S variant contained a disulfide bond. In contrast to the very stable disulfide bond found in the wild type, in AK2$^{C232S}$ and AK2$^{C40S}$, the disulfide bonds in AK2$^{C92S}$ and AK2$^{C92S,C232S}$ were very labile toward treatment with the reductant DTT. Although these data were obtained with purified proteins and not with unfolded import intermediates as they would be found during vectorial import into the IMS, our data indicate that in principle, a disulfide bond between C40 and C42 can be formed. This disulfide bond is very labile, and thus, it would be thermodynamically favorable to rearrange it to the C42–C92 disulfide. The *in vitro* data also again support the presence of a very stable native disulfide bond between C42 and C92.

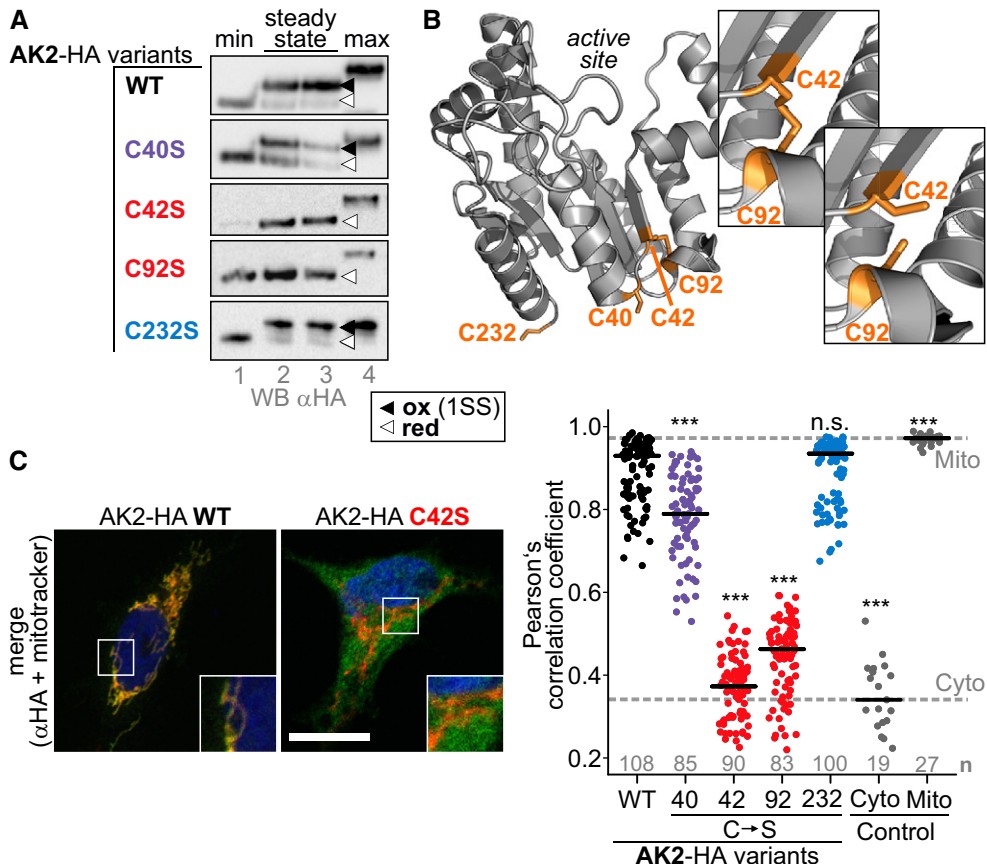

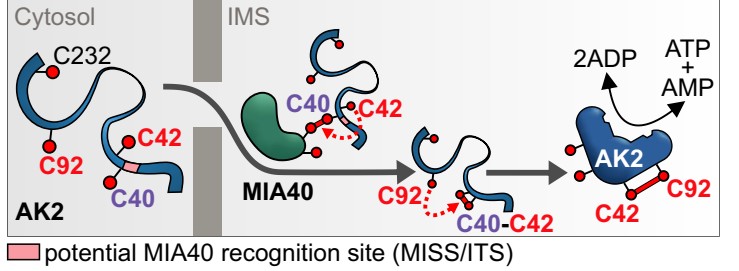

**Figure 2. Cysteine residues 42 and 94 are critical for mitochondrial accumulation of AK2.**

A  Influence of AK2 cysteines on the AK2 redox state. Cells expressing single cysteine to serine mutants of AK2-HA (isoform A) were lysed after NEM treatment and analyzed by inverse redox shift assay using mmPEG$_{12}$, SDS–PAGE, and immunoblot against the HA tag. AK2$^{C42S}$ and AK2$^{C92S}$ are present in their completely reduced state indicating presence of a disulfide bond between both cysteines.

B  Crystal structure of AK2 (pdb entry 2C9Y). In the crystal structure, a disulfide bond between C42 and C92 is visible with about 50% occupancy, while 50% of the Cys42/92 residues are in the reduced state. This disulfide bond is far away from the active site of AK2. The figure was created using PyMOL.

C  Influence of AK2 cysteines on cellular localization. Cells expressing single cysteine to serine mutants of AK2-HA (isoform A) were fixed and analyzed by immunofluorescence. The colocalization of AK2-HA variants with MitoTracker was quantified using CellProfiler (Pearson's correlation coefficient). C42 and C92 are critical for mitochondrial localization of AK2. The C40S mutant exhibits significantly decreased colocalization with mitochondria. A Kruskal–Wallis rank sum test/Dunn's test was performed. Numbers of analyzed cells are indicated in the plot. *** represents *P*-value < 0.001 compared to wild type. The black horizontal line represents the median. Data information: Scale bars: 10 μm.

D  Model for AK2 import and oxidative folding. See text for details.

Source data are available online for this figure.

Taken together, our data support a model in which an initial interaction between C40 and MIA40 drives import of AK2 into the IMS. Subsequent formation of a C40–C42 disulfide and later intramolecular rearrangement to the native C42–C92 disulfide result in mature AK2. We propose that this C40-dependent reaction sequence is important to accelerate folding and import of AK2 compared to a situation without C40 (and thus without isomerization, Fig 2D).

## AK2 import competes with proteasomal degradation

Diminished disulfide relay substrate levels are often a hallmark of MIA40-depletion (Hangen *et al*, 2015; Habich *et al*, 2019a). In line with this, AK2 was almost absent from cells when MIA40 was depleted by siRNA-mediated knockdown (Fig 3A, lanes 2, 3) or dominant-negative MIA40 variants were expressed ((Habich *et al*, 2019a), Appendix Fig S3A, lane 3). This was due to the short half-life of AK2 under these conditions as we demonstrated by pulse-chase experiments (Fig 3B).

MIA40 substrates that fail to become imported into the IMS are eventually degraded by the proteasome (Habich *et al*, 2019a; Mohanraj *et al*, 2019). To test whether this is also the case for AK2, we assessed the stability of an import-deficient cysteine variant of AK2 (AK2$^{C40S,C42S,C92S}$, localization see Appendix Fig S3B) in dependence of the proteasomal inhibitor MG132. Indeed, this AK2 variant was very unstable and could be stabilized by MG132 (Fig 3C). In line, steady-state levels of AK2$^{C40S,C42S,C92S}$-HA but also of AK2$^{WT}$-HA were increased upon MG132 incubation or siRNA-mediated depletion of proteasomal subunits PSMB5/C3 (Fig 3D and E). Our findings thus indicate that cytosolic accumulation of (wild type and mutant) AK2 is actively prevented by competition between import and quick proteasomal degradation.

## Dipeptidyl peptidase 9 targets AK2 for proteasomal degradation

Protein stability in the cytosol is often determined by the N-terminal amino acids of proteins (Varshavsky, 2017). We thus analyzed the N-terminus of wild-type AK2-HA after its immunoprecipitation by proteomic approaches (Fig 4A). We thereby found that the most N-terminal peptide of the AK2 protein existed in four different versions: the native N-terminus ("MAPS-"), an N-terminus, which lacked the methionine and likely was processed by methionine aminopeptidase MAP ("APS-"), and two N-termini, which lacked the three most N-terminal amino acids ("S-"). Of the latter, one variant was N-terminally acetylated while the other was not. Notably, assessment of AK2 conservation in 163 species revealed that the three most N-terminal amino acids including the proline are highly conserved (Fig 4A, logo plot) indicating that these amino acid residues might fulfill an evolutionary conserved role.

The identification of different AK2 N-termini might constitute the regulatory mechanism to control cytosolic stability of AK2. Given the fact that the IMS does not harbor the respective degradation machinery, a protein that escapes the cytosolic N-end rule machinery by becoming imported would be stable in the IMS. Which protease facilitates N-terminal processing of AK2? Members of the dipeptidyl peptidase 4 (DPP4) protease family possess the unusual capacity to cleave a post-proline bond to release dipeptides from the N-terminus of proteins (Wilson *et al*, 2016; Sato & Ogita, 2017). Two members of this family are present in the cytosol, DPP8 and DPP9 (Wilson *et al*, 2016).

We tested whether DPP8 and/or DPP9 affected stability of cytosolic AK2 (AK2$^{C40S,C42S,C92S}$-HA, Fig 4B and C). Indeed, siRNA-mediated depletion of DPP8 and/or DPP9 revealed a stabilization of this AK2 variant at steady state (Fig 4B). The stabilization was thereby most prominent when DPP9 was depleted but also increased further upon double depletion of both DPPs. Since DPP8 alone did not affect AK2$^{C40S,C42S,C92S}$-HA stability, we reasoned that DPP9 might be the main processing enzyme with DPP8 serving as backup. We complemented the siRNA-based experiment with an experiment in which we employed 1G244, a specific inhibitor of DPP8 and DPP9 (Wu *et al*, 2009; Fig 4C). We thereby found a strong stabilization of AK2$^{C40S,C42S,C92S}$-HA upon DPP8/9 inhibition. Our experiments so far were performed with an HA-tagged AK2 cysteine variant. Next, we repeated the experiments with endogenous AK2 (Fig 4D and E). We thereby found that also endogenous AK2 levels increased upon DPP9 depletion or inhibition of DPP8/9 activity. Thus, DPP9 indeed affects cellular AK2 levels.

To test whether DPP8/9 directly process AK2, we introduced mutations into the N-terminus of AK2$^{WT}$ or AK2$^{C40S,C42S,C92S}$ and analyzed the stability of the respective variants (Fig 4F and Appendix Fig S4A). The first mutation, replacement of alanine at position 2 by aspartate (A2D), hampers processing by MAP and as a consequence also the processing by DPP9. This AK2 variant would

---

**Figure 3. AK2 import competes with proteasomal degradation.**

A   Steady-state levels of endogenous AK2 in presence and absence of MIA40. 72 h before experiments, HeLa cells were transfected with control siRNA or siRNAs directed against MIA40 or ALR. Cell lysates were analyzed by immunoblot against the indicated proteins. MIA40 depletion led to a strong decrease in AK2 and ALR levels but not of the control proteins. ALR depletion did not affect AK2 levels. Reported values are the mean of 3 independent experiments; error bars represent $\pm$ SD. Student's *t*-test was performed. * represents $P < 0.05$.

B   Stability of endogenous AK2 in the presence and absence of MIA40. 72 h before experiments HeLa, cells were transfected with control siRNA or siRNAs directed against MIA40. Cells were pulse-labeled for 10 min with [$^{35}$S]-methionine and chased with cold methionine for the indicated times. The chase was stopped by cell lysis, and AK2 was isolated by denaturing immunoprecipitation. Eluates were analyzed by SDS–PAGE and autoradiography. Stability of endogenous AK2 is significantly decreased upon depletion of MIA40. Reported values are the mean of 3 independent experiments; error bars represent $\pm$ SD. Student's *t*-test was performed; * represents *P*-value $< 0.05$ compared to wild type.

C   Stability of cytosolic AK2-HA cysteine variant. Experiment was performed as in (B) except that cells expressing AK2$^{C40S,C42S,C92S}$-HA were analyzed by IP against HA. AK2$^{C40S,C42S,C92S}$-HA is unstable but becomes stabilized by application of the proteasomal inhibitor MG132. Reported values are the mean of 3 independent experiments; error bars represent $\pm$ SD. Student's *t*-test was performed. * represents $P < 0.05$.

D   Steady-state levels of AK2-HA cysteine variants. AK2$^{C40S,C42S,C92S}$-HA and AK2$^{WT}$-HA were expressed in the presence or absence of proteasomal inhibitor MG132. Protein levels at steady state were analyzed by immunoblot. All variants are stabilized by proteasomal inhibition. Reported values are the mean of 3 (AK2$^{WT}$-HA) or 5 (AK2$^{C40S,C42S,C92S}$-HA) independent experiments; error bars represent $\pm$ SD. ANOVA/Tukey's *post hoc* test was performed. *** represents $P < 0.001$.

E   Steady-state levels of AK2-HA cysteine variant. 72 h before experiments, HEK293 cells were transfected with control siRNA or siRNAs directed against the proteasomal subunits PSMB3 and PSMC5. AK2$^{C40S,C42S,C92S}$-HA was stabilized by decreased proteasome levels. Reported values are the mean of 4 independent experiments; error bars represent $\pm$ SD. Student's t-test was performed. * represents $P < 0.05$.

Source data are available online for this figure.

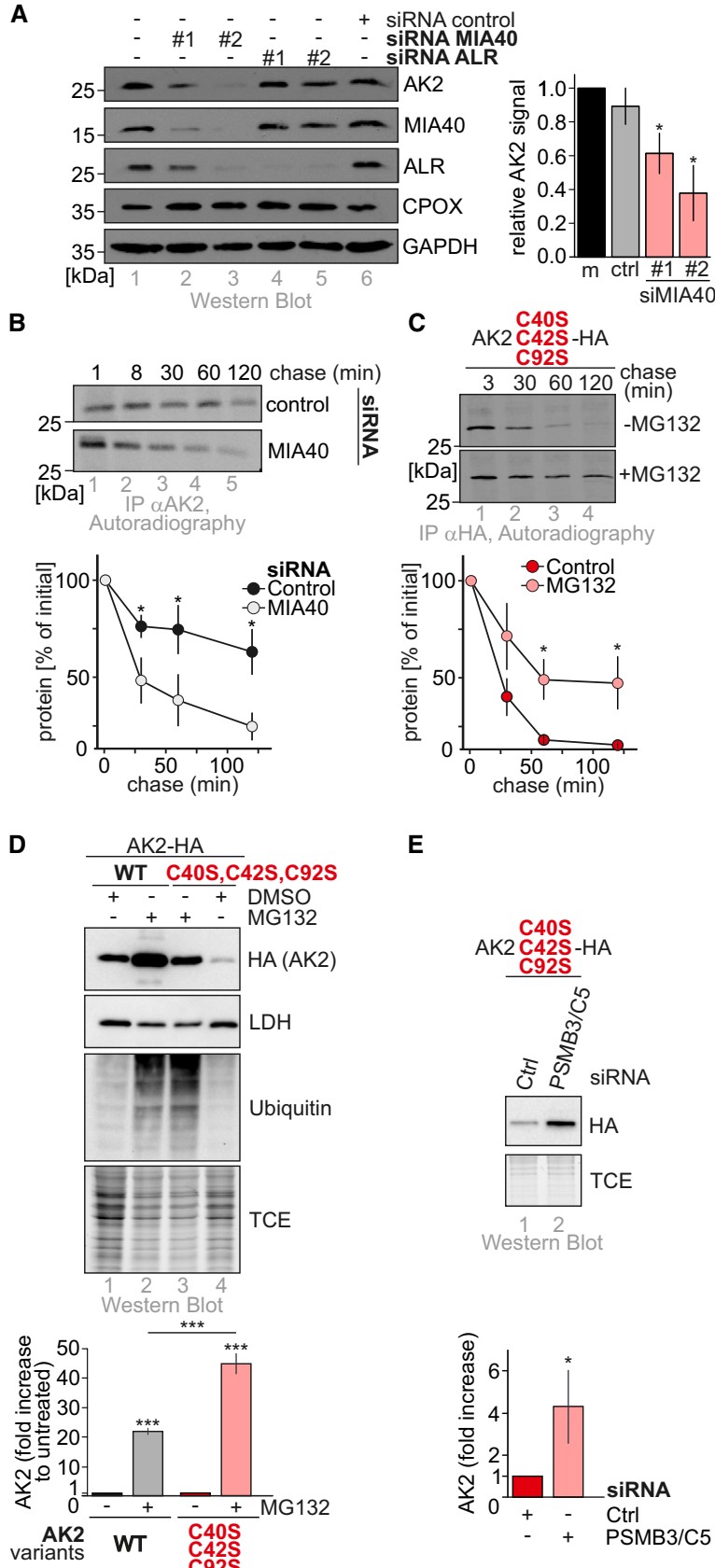

**Figure 3.**

carry a methionine at its N-terminus (Appendix Fig S4A). As second mutation, we replaced serine at position 4 by proline, which prevents cleavage by DPP9 and thus leaves alanine as the neo-N-terminus of this AK2-variant (S4P). When we analyzed the levels of these variants either in the context of wild-type AK2 or the C40S, C42S,C92S variant of AK2, we found a striking stabilization of AK2 upon modification of its N-terminus (Fig 4F). This effect was not further increased by application of 1G244 or siRNA-mediated depletion of DPP9 indicating that both, the A2D and the S4P mutation, prevented processing in the first place (Fig 4F and Appendix Fig S4B). Moreover, incubation with MG132 did not additionally stabilize AK2$^{A2D,C40S,C42S,C92S}$-HA and AK2$^{S4P,C40S,C42S,C92S}$-HA (Appendix Fig S4C). This indicates that AK2 is not targeted for proteasomal degradation in the absence of processing by DPP9. Lastly, we tested the effect of replacing serine at position 4 by either valine or glycine (S4V, S4G) (Appendix Fig S4A and D). In both mutants, DPP9 cleavage is allowed but instead of an N-terminal serine, AK2 will then carry valine or glycine as neo-N-terminus. Compared to serine, these two amino acids should provide AK2 with increased resistance to degradation. Indeed, both AK2 variants showed increased stability (Appendix Fig S4D).

Taken together, DPP9 processes the N-terminus of AK2 in the cytosol. This targets AK2 for proteasomal degradation. Once AK2 reaches the IMS, it is protected from degradation.

### DPP9 processing prevents cytosolic AK2 accumulation

Why is AK2 specifically targeted for rapid cytosolic degradation? One reason might be that upon cytosolic accumulation, AK2 could be capable of folding and acquiring enzymatic activity. We first investigated whether AK2 indeed accumulates in the cytosol when DPP9 processing does not take place. Using cell fractionation approaches, we found that a small amount of endogenous AK2 accumulated in the cytosolic fraction upon treatment with DPP9 inhibitor, while the IMS proteins CPOX and cytochrome *c* remained completely in the mitochondrial fraction (Fig 5A), thus excluding contamination. In agreement, by immunofluorescence analyses, we found that the DPP9-insensitive variant AK2$^{S4P}$ that otherwise can become normally imported into mitochondria was partially localized in the cytosol. AK2$^{WT}$ fully localized to mitochondria (Fig 5B and Appendix Fig S5A). Lastly, AK2$^{C40S}$, the variant that can become oxidized and imported into mitochondria albeit with slower kinetics compared to wild-type AK2, accumulated more strongly in the cytosol when proteasomal degradation was inhibited (Fig 5C and Appendix Fig S5B and C). In summary, inhibition of AK2 processing by DPP9 results in accumulation of a cytosolic AK2 species and the effects are more pronounced when using AK2 variants with delayed import kinetics.

### Cytosolic AK2 is stable and enzymatically active

Mitochondrial proteins are imported in an unfolded state. "Mislocalization" of AK2 might thus result from premature folding in the cytosol. AK2 is one of the few disulfide relay substrates with inherent enzymatic activity. We thus next assessed whether cytosolic AK2 acquires enzymatic activity (Fig 6). We first established two activity assays for AK2 function, one with respect to its substrate AMP and one toward its substrate ADP (Fig 6A and B, and Appendix Fig S6A–C). Unfortunately, all AK2 variants still containing C232 tended to form non-physiological disulfide-bonded dimers involving this cysteine (Appendix Fig S6D). This was not the case for AK2$^{WT}$. To avoid the comparison of dimeric with monomeric AK2 variants, we decided to employ instead of the wild type the "wild-type-like" AK2 variant, AK2$^{C40S,C232S}$ that contains the C42-C92 structural disulfide bond but lacks the other cysteines. First we confirmed that AK2$^{C40S,C232S}$ and AK2$^{WT}$ had similar activities

---

**Figure 4. AK2 is target of the cytosolic dipeptidyl peptidases DPP8 and DPP9.**

A   The N-terminus of AK2. Proteomic analysis of wild-type AK2 immunoprecipitated from HEK293 cells revealed the presence of four different N-termini: "MAPS-," "APS-," "S-," and "acetylated S-." The processing from "MAPS-" to "APS-" is likely caused by methionine amino peptidase (MAP). The second processing step might be caused by the cytosolic DPP4 protease family members DPP8 and 9 that are known to cut at the N-terminus peptides of two amino acids length preferably after prolines. The N-terminus of AK2 is highly conserved as a sequence analysis of AK2 homologues from 163 species revealed (logo plot).

B   Steady-state levels of AK2$^{C40S,C42S,C92S}$-HA in presence and absence of DPP8 and DPP9. 72 h before experiments, HEK293 cells stably expressing AK2$^{C40S,C42S,C92S}$-HA were transfected with control siRNA or siRNAs directed against DPP8 or/and DPP9. Cell lysates were analyzed by immunoblot against the indicated proteins. DPP9 depletion led to an increase in AK2$^{C40S,C42S,C92S}$-HA levels, while DPP8 depletion did not. Combined depletion of DPP8 and DPP9 increased AK2$^{C40S,C42S,C92S}$-HA levels even further. This indicates that DPP9 is the main peptidase and DPP8 can serve as its backup. White and black arrow heads indicate the endogenous and HA-tagged AK2, respectively. The gray arrow head indicates the DPP9 band. Reported values are the mean of 3 independent experiments; error bars represent ± SD. ANOVA/Tukey's *post hoc* test was performed. * represents $P < 0.05$, ## represent $P < 0.01$, and ***.### represent $P < 0.001$.

C   Steady-state levels of AK2$^{C40S,C42S,C92S}$-HA upon incubation with the DPP9 inhibitor 1G244 (Wu *et al*, 2009). As (B) except that cells were treated for 16 h with inhibitor or DMSO as control. AK2$^{C40S,C42S,C92S}$-HA levels were strongly increased upon DPP8/9 inhibition. Reported values are the mean of 4 independent experiments; error bars represent ± SD. Student's *t*-test was performed. *** represents $P < 0.001$.

D   Steady-state levels of endogenous AK2 in presence and absence of DPP9 and MIA40. 72 h before experiments HeLa cells were transfected with control siRNA or siRNAs directed against MIA40 and/or DPP9. Cell lysates were analyzed by immunoblot against the indicated proteins. If import of endogenous AK2 was prevented by depletion of MIA40, then concomitant depletion of DPPs increased its levels. Strikingly, depletion of DPP9 alone also increased the levels of endogenous AK2. Reported values are the mean of 3 independent experiments; error bars represent ± SD. ANOVA/Tukey's *post hoc* test was performed. ** represents $P < 0.01$.

E   Steady-state levels of endogenous AK2 upon incubation with the DPP9 inhibitor 1G244. As (D) except that cells were treated for 3 or 5 days with inhibitor or DMSO as control. Endogenous AK2 levels were increased upon DPP8/9 inhibition. Reported values are the mean of 5 (3 days) and 6 (5 days) independent experiments; error bars represent ± SD. Student's *t*-test was performed. * represents $P < 0.05$, and *** represents $P < 0.001$.

F   Steady-state levels of AK2-HA variants with mutated N-termini upon incubation with either 1G244 or vehicle control. HEK293 cells stably expressing AK2-HA variants were lysed and subjected to immunoblot against the indicated proteins. Levels of DPP8/9-insensitive AK2 variants are strongly increased. Reported values are the mean of 4 independent experiments; error bars represent ± SD. ANOVA/Tukey's *post hoc* test was performed. ** represents $P < 0.01$ and ###.*** represent $P < 0.001$. Asterisk denominate statistical comparison with respective control, and hash tag represents comparison to N-WT.

Source data are available online for this figure.

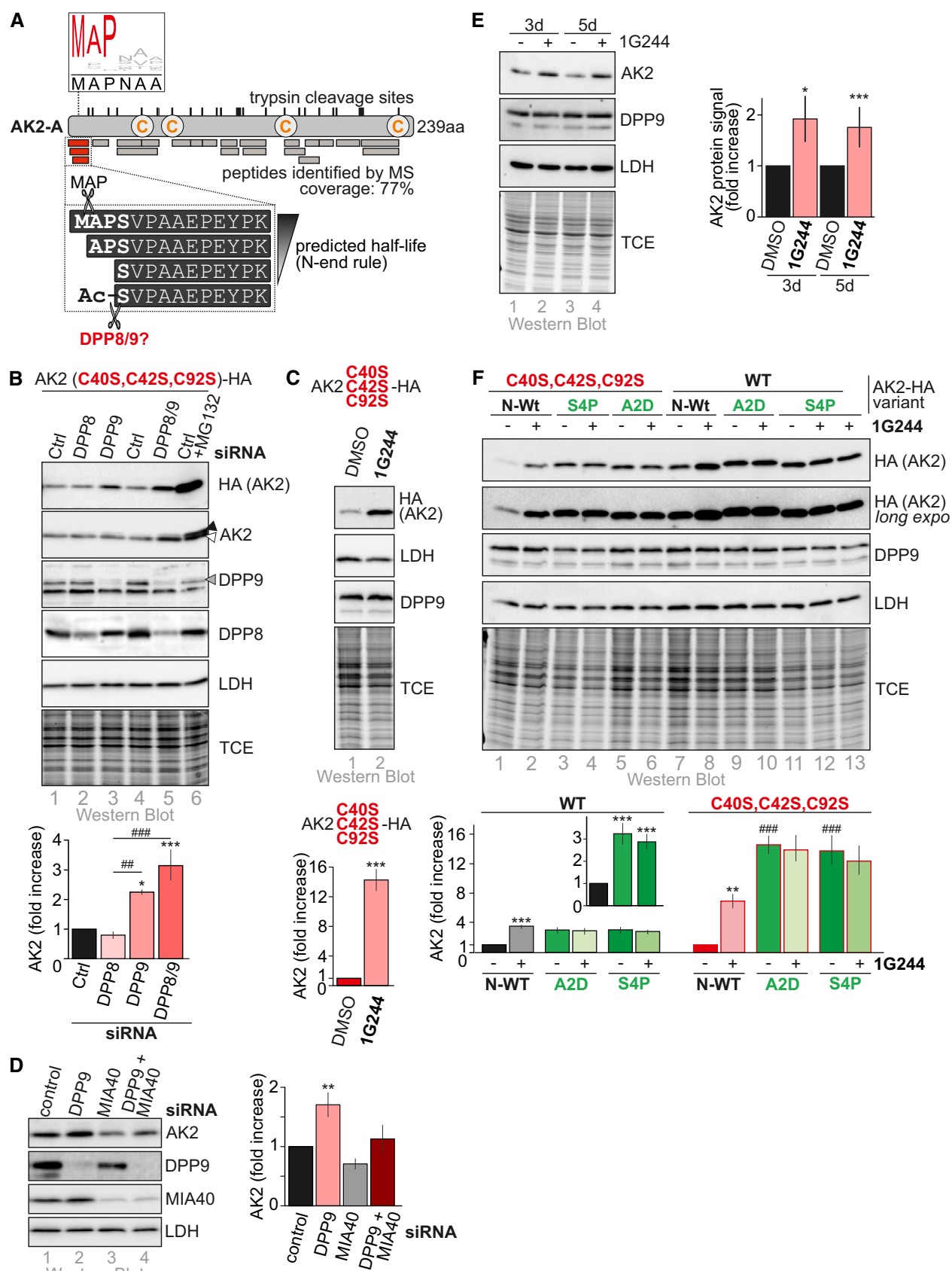

**Figure 4.**

toward AMP and ADP as substrates (Appendix Fig S6E). We then compared $AK2^{C40S,C232S}$ (only containing the disulfide-forming C42 and C92) with a cysteine-less AK2 variant. We observed that purified AK2 exhibited typical substrate inhibition curves as also observed for other AKs (Liang *et al*, 1991; Sinev *et al*, 1996; Fig 6C and D). We observed maximal activity for 40 μM AMP and 1.5 mM ADP, respectively. With increasing AMP concentrations, AK2 activity dramatically decreased switching off the enzyme almost completely at about 0.5–1 mM AMP. For ADP, this decrease in activity featured a more gradual decline. Importantly, irrespective of the presence of the structural disulfide bond in AK2, enzyme kinetic profiles were the same (compare $AK2^{C40S,C232S}$ and $AK2^{C40S,C42S,C92S,C232S}$, Fig 6C and D). Thus, the presence of the structural disulfide bond had no influence on AK2 enzymatic activity.

Next, we developed an assay that allowed the determination of AK2 activity from samples isolated from HEK293 cells (Fig 6E and Appendix Fig S6F). Cells stably expressing different AK2-HA variants were lysed using a mild detergent. Subsequently, AK2 was isolated using HA antibodies. With the AK2-HA variants still bound to the beads, the AK2 activity assay with the substrate ADP was performed. The enzymatic inactive K28Q variant of AK2 (Reinstein *et al*, 1990) thereby served as negative control to ensure that no endogenous AK activity was precipitated from cells. In the absence of DPP8/9 inhibitor, we found that mitochondrial AK2 variants show similar activity irrespective whether they are DPP8/9 insensitive. The cytosolic $AK2^{C40S,C42S,C92S}$ showed very low activity; yet this activity was clearly higher than the one of the K28Q control. The DPP8/9-insensitive variants $AK2^{A2D,C40S,C42S,C92S}$ and $AK2^{S4P,C40S,C42S,C92S}$ exhibited activity almost as strong as wild-type AK2. Upon pretreatment of cells with DPP8/9 inhibitor, especially the activity of $AK2^{C40S,C42S,C92S}$ increased almost to wild-type levels. When we normalized these activities to precipitated protein levels, we found specific activities of cytosolic and IMS-AK2 to be relatively similar (Appendix Fig S6G). Taken together, AK2 can accumulate and fold in the cytosol and acquire activity. Preventing DPP9 processing and thereby proteasomal degradation increases the amount of cytosolic AK2 levels and activity.

Cytosolic AK2 is not oxidatively folded. This raises the question to which extent the native C42-C92 disulfide bond affects AK2 stability. To test this, we determined the melting temperature of AK variants (Appendix Fig S6H) and found it to be 58.4°C for $AK2^{C40S,C232S}$ and 45.5°C for a disulfide-free mutant. With this, the disulfide-free AK2 has the same stability against thermal unfolding as the cytosolic adenylate kinase, AK1 (Appendix Fig S6H). Thus, the disulfide-free (cytosolic) AK2 is stable and AK2 containing the C42–C92 disulfide bond has increased stability. In the light of recent findings on potentially increased temperatures in the IMS (Chretien *et al*, 2018), this implies that disulfide formation in AK2 not only serves in allowing mitochondrial accumulation but also increases protein stability to allow function in the IMS.

To test whether cytosolic AK2 activity impacted cellular fitness, we expressed different AK2 variants in HEK293 cells, exposed them to different metabolic conditions (galactose, glucose) and scored their proliferation in comparison to Mock and the dominant-negative $MIA40^{C53S}$ variant as control (Appendix Fig S6I). While $MIA40^{C53S}$ expression led to a clear growth defect, expression of the different AK2 variants had no growth defect, indicating that in this setting cytosolic AK2 did not impact cell viability.

Collectively, our data support a model, in which DPP9 processes the N-terminus of AK2 in the cytosol to promote proteasomal degradation and to ensure that AK2 activity is absent from the cytosol under normal conditions. Importantly, in the IMS, N-terminally processed AK2 is protected from proteasomal degradation. Inhibition of DPP9 results in increased cellular AK2 levels and cytosolic AK2 activity. While we achieved inhibition by introducing mutations or by chemical inhibitors, DPP9 activity might also be lowered by transcriptional regulation or might be inhibited by posttranslational modifications, although very little is known about modes of DPP9 regulation (Fig 6F).

## DPP9 targets additional mitochondrial proteins

Our data support a critical role of N-terminal processing by DPP8/9 for the regulation of AK2 levels and cellular localization. To test whether other mitochondrial proteins are targeted by DPP8/9, we screened the MitoCarta2.0, which contains proteins with a strong support for mitochondrial localization (Calvo *et al*, 2016) for DPP8/9 recognition and processing motifs. These motifs include MXPY and MPY (with X not D/E, and Y not P) (Wilson *et al*, 2013, 2016; Zhang *et al*, 2015; Fig 7A, Dataset EV1). We thereby identified 107 potential mitochondrial DPP8/9 substrates. Interestingly, IMS proteins and disulfide relay substrates appeared to be enriched among the potential DPP8/9 targets.

Among the potential DPP8/9 substrates were 13 disulfide relay substrates including AK2 (Fig 7B). Their N-termini are often conserved, indicating that DPP8/9 cleavage might be an evolutionary widely applied regulatory pathway regarding substrates and species. Moreover, analysis of the mouse N-terminome (Calvo *et al*, 2017) revealed two disulfide relay substrates, COX17 and NDUFA8 that as mature proteins carry N-termini, as they would occur after DPP8/9 cleavage. Disulfide relay substrates often are not detected in proteomic approaches, and also in this proteomics approach, no other disulfide relay substrates were identified.

To complement our bioinformatics analysis, we tested effects of DPP8/9 inhibition on the disulfide relay substrates, NDUFB10, NDUFA8, NDUFS5, COA6, and CHCHD2 (Fig 7C). We cultured in the presence or absence of 1G244 HEK293 cells that stably and inducibly express HA-tagged variants of these proteins. We thereby found that indeed all proteins became stabilized upon 1G244 treatment. Conversely, the disulfide relay substrate TIMM9 that does not qualify as DPP8/9 substrate did also not become stabilized upon 1G244 treatment (Fig 7C).

We had previously identified a disease variant in NDUFB10, $NDUFB10^{C107S}$ that was impaired in mitochondrial import (Friederich *et al*, 2017). The majority of the mutant protein was degraded by the proteasome, and consequently proteasomal inhibition stabilized $NDUFB10^{C107S}$ (Habich *et al*, 2019a). Proteasomal inhibition has recently been proposed as mode to rescue the mitochondrial accumulation of a disease variant of COA7 that was likewise impaired in its mitochondrial import (Mohanraj *et al*, 2019). We tested whether DPP9 inhibition would allow a similar rescue for $NDUFB10^{C107S}$ (Fig 7D and E). To this end, we cultured HEK293 cells expressing either $NDUFB10^{WT}$-HA or $NDUFB10^{C107S}$-HA in the presence or absence of 1G244. We thereby found that indeed $NDUFB10^{C107S}$-HA became stabilized upon 1G244 treatment (Fig 7D). Remarkably, the fraction of the protein that accumulated

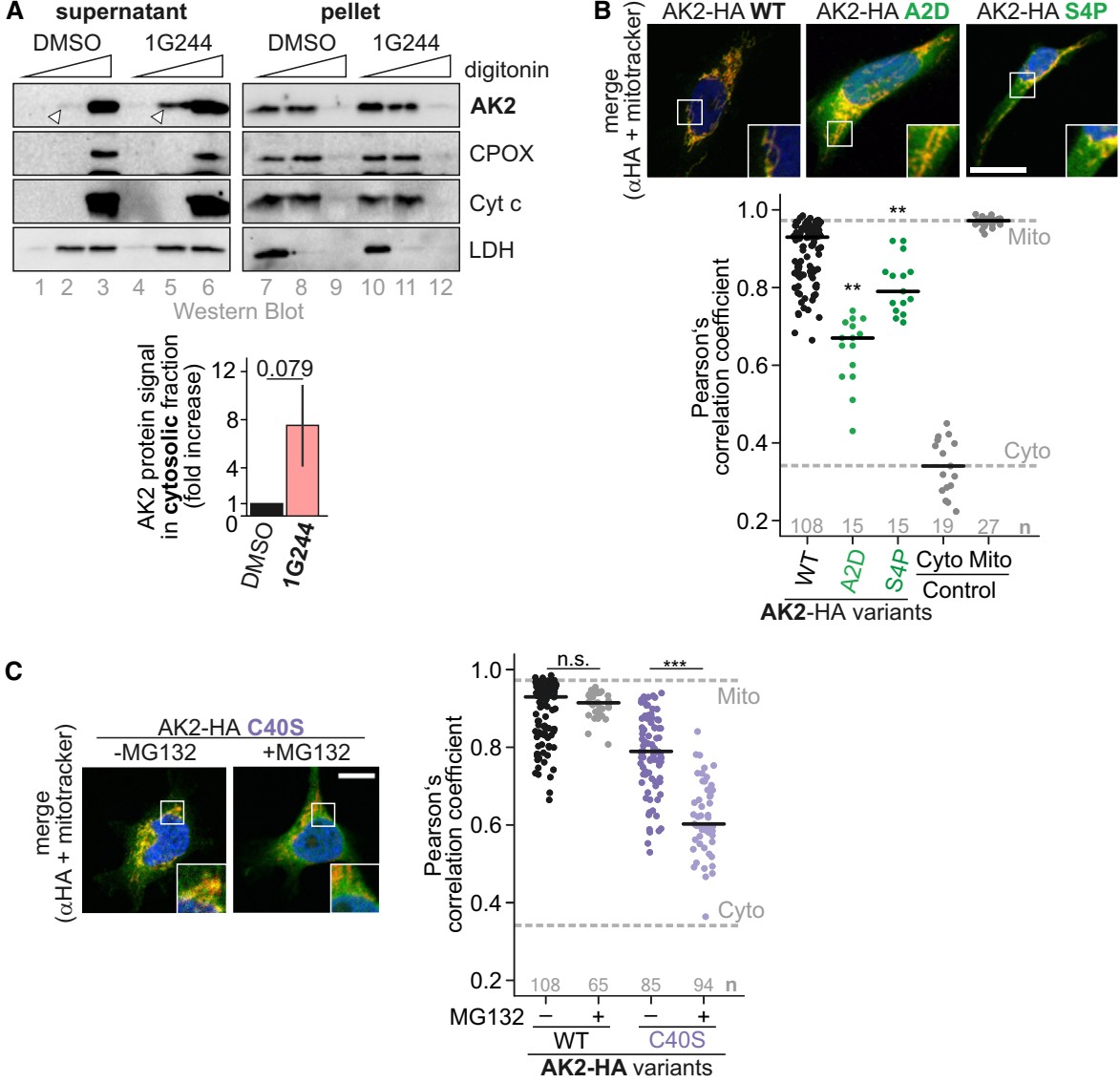

**Figure 5. Cytosolic AK2 accumulates when DPP9 processing is prevented.**

A  Cell fractionation upon inhibition of DPP9 activity. HeLa cells were incubated for 4 days with the DPP9 inhibitor 1G244 or as control with DMSO. Cells were subjected to fractionation into "cytosol" (supernatant) and "mitochondria" (pellet) by differential digitonin lysis and centrifugation. The IMS marker proteins CPOX and cytochrome *c* were exclusively localized to the pellet fraction at one specific digitonin concentration (lanes 2, 5, 8, 11) irrespective of 1G244 treatment. Likewise, the cytosolic marker protein LDH localized under all conditions to the same fractions. Conversely, endogenous AK2 appeared at this digitonin concentration in the "cytosol" fraction but only upon 1G244 treatment (compare lanes 2 and 5). Thus, inhibition of DPP9 results in appearance of AK2 in the cytosol. Quantification shows loading-control adjusted AK2-signals of 1G244-treated "cytosol" fraction (lane 5) over DMSO-treated "cytosol" fraction (lane 2). White arrow head indicates the endogenous AK2 appearing in the cytosolic fraction upon 1G244 treatment. Reported values are the mean of 3 independent experiments; error bars represent ± SD. Student's *t*-test was performed.

B  Localization of DPP9-insensitive AK2-HA variants. Experiment was performed as described in Fig 2C. Compared to the wild type, DPP9-insensitive and import-competent AK2-HA variants colocalized with cytosolic markers indicating partial cytosolic localization. The colocalization of AK2-HA variants with MitoTracker was quantified using Fiji (Pearson's correlation coefficient). A one-way ANOVA with *post hoc* Tukey test was performed. Numbers of analyzed cells are indicated in the plot. ** represents $P < 0.01$. Data shown in Figs 2C and 5B, and Appendix Figs S2 and S5 are derived from the same experiments. The black horizontal line represents the median. Data information: Scale bars: 10 μm.

C  Localization of AK2[C40S]-HA upon proteasomal inhibition. Experiment was performed as described in Fig 2C. Upon MG132 treatment, colocalization of AK2[C40S]-HA with the cytosolic marker increased. A one-way ANOVA with *post hoc* Tukey test was performed. Numbers of analyzed cells are indicated in the plot. *** represents $P < 0.001$. The black horizontal line represents the median. Data information: Scale bars: 10 μm.

Source data are available online for this figure.

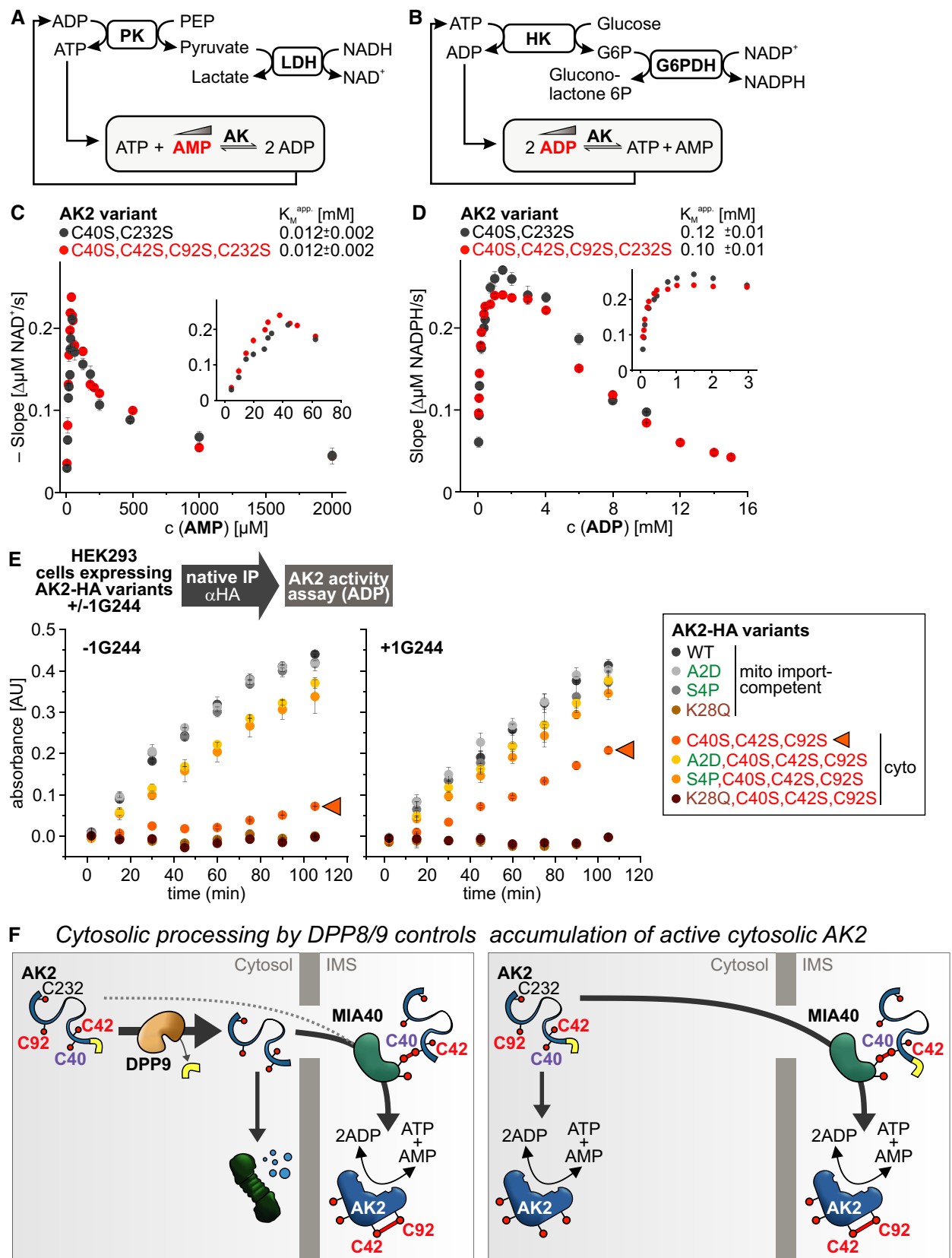

**Figure 6.**

◀

**Figure 6. Cytosolic AK2 is active.**

A  AK2 activity assay with varying AMP concentrations. Pyruvate kinase (PK) and lactate dehydrogenase (LDH) thereby couple the oxidation of NADH to the interconversion of adenine nucleotides. The concentration of ATP is maintained constant, whereas the concentration of AMP is titrated.

B  AK2 activity assay with varying ADP concentrations. Hexokinase (HK) and glucose-6-phosphate dehydrogenase (G6PDH) thereby couple the reduction of NADP$^+$ to the interconversion of adenine nucleotides. The concentration of ADP is titrated.

C  AK2 activity in dependence of its substrate AMP. Heterologously expressed and purified AK2 variants exhibited maximal activity at around 40 μM AMP with an apparent $K_M$ of 0.012 mM AMP. The profiles for both AK2$^{C40S,C232S}$ and AK2$^{C40S,C42S,C92S,C232S}$ are superimposable. Reported values are the mean of at least 3 independent experiments; error bars represent ± SD.

D  AK2 activity in dependence of its substrate ADP. Maximal activity was observed at around 1–2 mM ADP with an apparent $K_M$ of 0.12 mM and 0.1 mM ADP for AK2$^{C40S,C232S}$ and AK2$^{C40S,C42S,C92S,C232S}$, respectively. The profiles for both AK2$^{C40S,C232S}$ and AK2$^{C40S,C42S,C92S,C232S}$ are thus highly similar. Reported values are the mean of at least 3 independent experiments; error bars represent ± SD.

E  Activity assay for AK2-HA variants isolated from HEK293 cells. Indicated AK2-HA variants were isolated by native immunoprecipitation from HEK293 cells stably expressing these variants. Then, the AK2 activity assay described in (B) was performed. While import-competent AK2 variants exhibited full activity, a cytosolic AK2 variant with the native N-terminus showed very low activity. This activity strongly increased upon incubation with the DPP9 inhibitor 1G244 or upon using DPP9-insensitive cytosolic AK2 variants. The enzymatic inactive K28Q variants served as negative control indicating that only AK2-HA-dependent activity ways measured by our assay. Reported values are the mean of 2 independent experiments; error bars represent ± SD.

F  Model for the cytosolic quality control pathway controlling cellular AK2 level and its localization. *Details see text.*

Source data are available online for this figure.

in mitochondria upon 1G244 treatment also increased, indicating that DPP8/9 inhibition indeed provided the additional time required for NDUFB10$^{C107S}$ to reach at least partially the IMS (Fig 7E). Notably, also NDUFB10$^{WT}$ became stabilized and its mitochondrial levels increased strengthening the notion that NDUFB10 is a target of DPP9. Collectively, our data indicate that DPP8/9 processing might be a widely applied mechanism to influence mitochondrial protein import.

# Discussion

### Cytosolic processing by DPP8/9 controls cellular AK2 levels and dual AK2 localization

In this study, we identified a novel import regulation and quality control pathway for mitochondrial proteins. This regulation relies on cytosolic processing by DPP8/9. We propose that this DPP8/9 processing serves as a timer that prevents aberrant cytosolic accumulation of mitochondrial precursors, regulates cellular protein levels, and enables rapid control of dual localization.

Our findings suggest that the DPP8/9 quality control pathway employs the N-end-rule degradation pathways to modulate the dwelling time of precursors in the cytosol. Processing of human AK2, for example, exposes a serine residue. An N-terminal serine residue targets cytosolic proteins to the Ac/N-end rule pathway (Varshavsky, 2017; Nguyen *et al*, 2018). In this pathway, the N-terminal amino group becomes acetylated by N-acetyl transferases, which generates recognizable degrons. In line, we identified an N-acetylated N-terminal peptide in our proteomic approach (Fig 4A) and found AK2 variants with mutated serine 4 to be more stable (Appendix Fig S4A and D). Notably, while the "MAP" motif in AK2 is highly conserved, the amino acid residue at position 4 is not. Besides serine, e.g., asparagine and arginine residues are present at this position. These (neo-) N-terminal residues target AK2 to different branches of the N-end rule pathway. For example, an N-terminal asparagine would thereby become deaminated to an aspartate residue. For bovine AK2 (containing an asparagine at position 4), this was indeed observed. In an analysis of its N-terminus, an

aspartate was observed at position 4 for some molecules in line with a preceding deamination step (Schlauderer & Schulz, 1996).

AK2 was in a previous high-throughput study identified as potential target for both DPP8 and DPP9 (Wilson *et al*, 2013). We demonstrate that DPP8 and DPP9 processing of AK2 indeed occurs in cells and that this processing regulates AK2 levels and localization. DPP8-dependent processing became however apparent only in the absence of DPP9 indicating that at least in HEK293 cells DPP9 is the dominant cytosolic protease of the two (Fig 4B). This is in line with the high sequence similarity, structure, and determined cleavage motif for DPP8 and DPP9 (Ross *et al*, 2018). It also indicates the potential for specific regulation by differential expression or post-translational modifications of both proteases.

What are the consequences of differential DPP8/9 activities for AK2? We demonstrated that absence of AK2 processing results in increased cellular levels of AK2 and in the accumulation of folded, active, cytosolic AK2 (Figs 5 and 6E). AK2 activity is linked to metabolic signaling (Dzeja *et al*, 2004, 2011; Dzeja & Terzic, 2009), to maintenance of OXPHOS efficiency (Burkart *et al*, 2011; Six *et al*, 2015), and to induction of the ER unfolded protein response (Burkart *et al*, 2011) as well as apoptosis (Single *et al*, 1998; Kohler *et al*, 1999; Lee *et al*, 2007). Control of DPP8/9 activity might thus serve in modulating AK2 function, e.g., during differentiation or mitochondrial biogenesis. Indeed, using a murine primary B-cell differentiation model, we found that DPP8/9 impacts AK2 levels (Fig 4G). AK2 has together with the DUSP26 phosphatase been reported to play a role in regulating cell proliferation and activation of apoptosis (Kim *et al*, 2014). This would require cytosolic stabilization of AK2. Attenuation of DPP8/9 processing would provide an explanation on how AK2 could remain stable in the cytosol to fulfill these functions.

Cytosolic AK2 has the same enzymatic properties as its IMS counterpart, including the stringent regulation by AMP inhibition in a range that might become relevant under starvation conditions (Fig 6C and D; Soboll *et al*, 1978, 1980). AK2 activity is however considerably lower than the one of its cytosolic counterpart AK1 (Appendix Fig S7A and B) indicating that cytosolic AK2 activity becomes relevant only in cells with low or absent AK1 activity. Cells of the hematopoietic system lack AK1 activity, explaining the

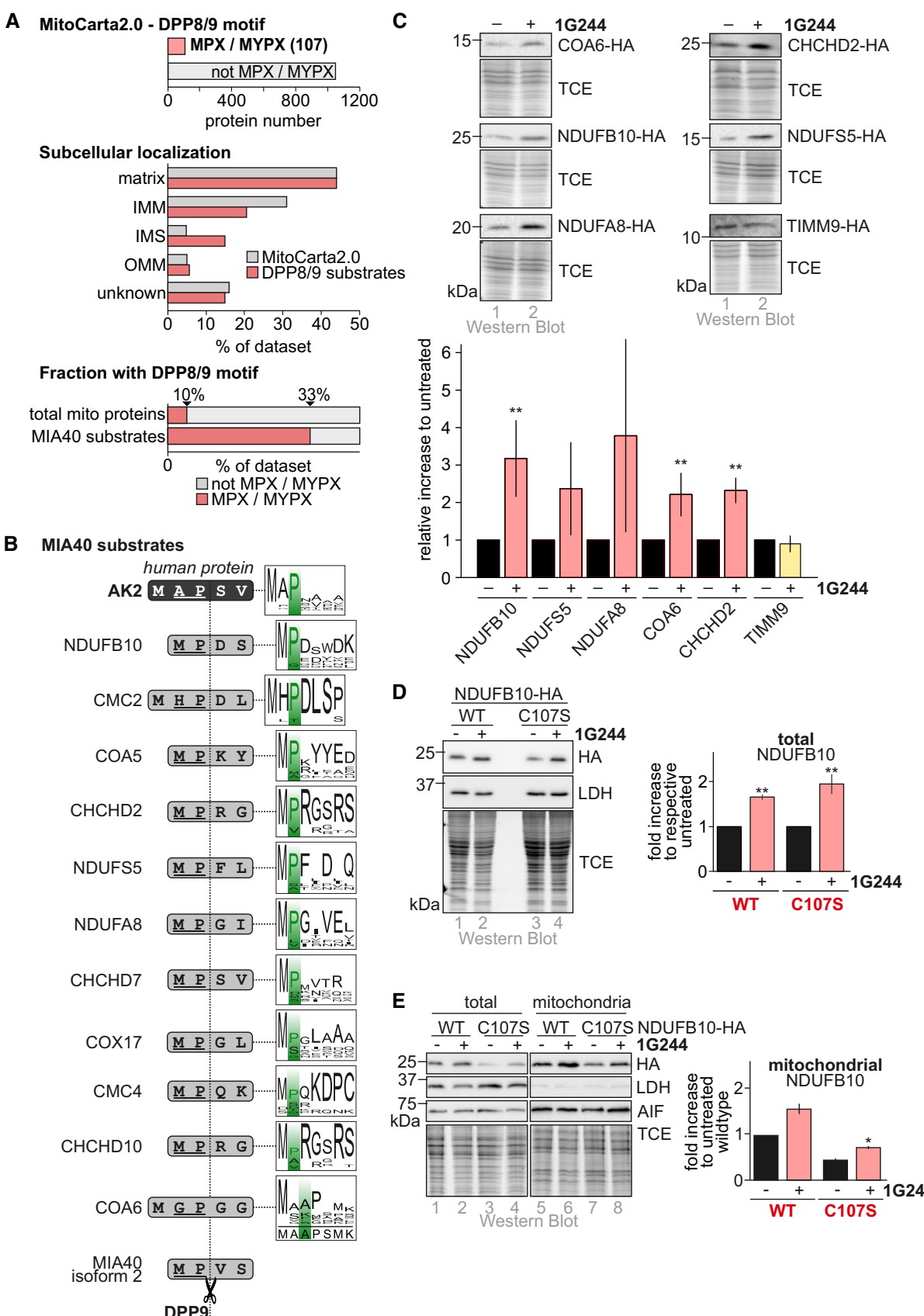

**Figure 7.**

◄

**Figure 7.  DPP9 controls levels and localization for multiple mitochondrial proteins.**

A   *In silico* assessment of the mitochondrial proteome ("MitoCarta 2.0," (Calvo *et al*, 2016)). The MitoCarta was searched for the following DPP9 recognition motifs: MPX (X not P) or MYPX (Y not D, X not P). More than 100 MitoCarta proteins fulfill this criterion. Of those, many do not carry an N-terminal mitochondrial targeting sequence (MTS) but instead internal or C-terminal targeting signals. Proteins found in IMS and IMM appear to be enriched.

B   *In silico* assessment of N-termini of human disulfide relay substrates. Approximately 40–50 substrates of the human disulfide relay have been described (Habich *et al*, 2019b). Of those, 13 contain conserved DPP8/9 processing sites at their N-terminus. For COX17 and NDUFA8, the predicted neo-N-termini "GL-" (COX17) and "GI-" (NDUFA8) after DPP9 processing were identified in the mouse N-terminome (Calvo *et al*, 2017). Logo plots represent alignments over 163 different species for AK2 and 11 different species for all other proteins. For many disulfide relay substrates, the cleavage sites for DPP8/9 are conserved.

C   Steady-state levels of NDUFB10, NDUFS5, NDUFA8, COA6, CHCHD2, and TIMM9 upon incubation with the DPP9 inhibitor 1G244. Cells were treated for 24 h with inhibitor or DMSO as control and were then analyzed by immunoblot against the indicated proteins. Proteins except TIMM9 are stabilized upon DPP9 inhibition. Reported values are the mean of 3–5 independent experiments; error bars represent ± SD. Student's *t*-test was performed. ** represents $P < 0.01$.

D   Steady-state levels of NDUFB10 variants upon incubation with the DPP9 inhibitor 1G244. Cells were treated for 24 h with inhibitor or DMSO as control and were then analyzed by immunoblot against the indicated proteins. NDUFB10$^{C107S}$ is stabilized upon DPP9 inhibition. Reported values are the mean of 4 independent experiments; error bars represent ± SD. Student's *t*-test was performed for the respective control and treatment samples. ** represents $P < 0.01$.

E   Localization of NDUFB10 variants upon 1G244 treatment. Experiment was performed as in (C) except that mitochondria were isolated and the fractions analyzed by immunoblot against the indicated proteins. Mitochondria contain increased NDUFB10$^{C107S}$ levels after DPP9 inhibition. Reported values are the mean of 2 independent experiments; error bars represent ± SD. Student's *t*-test was performed. * represents $P < 0.05$.

Source data are available online for this figure.

immune phenotype of AK2 patients (Lagresle-Peyrou *et al*, 2009; Pannicke *et al*, 2009), and it is tempting to speculate that here DPP9 processing might affect localization and cellular levels of AK2. This would also be in line with gene expression data (Bagger *et al*, 2019). Granulocyte/monocyte progenitors can differentiate into either monocytes or myelocytes. DPP9 transcripts were maintained at low levels during differentiation into early promyelocytes, while they were kept at higher levels during differentiation into monocytes (Appendix Fig S7C and Bagger *et al*, 2019). AK2 expression was inversely regulated which might point to increased AK2 levels and a cytosolic AK2 fraction in the case of differentiation toward myelocytes. Likewise, in bone marrow plasma cells, a striking inverse correlation of DPP8/9 expression and expression of their substrates can be observed (Appendix Fig S7D).

**DPP8/9 as general regulator of mitochondrial biogenesis and cellular protein localizations**

The role of DPP8/9 processing for mitochondrial biogenesis extends beyond its role in AK2 processing. Our *in silico* predictions show that more than 100 MitoCarta2.0 proteins contain the DPP8/9 processing site and might thus be potential targets of these peptidases (Fig 7). These predictions are supported by protein N-terminome data (Calvo *et al*, 2017). The proteins COX17 and NDUFA8 were thereby identified with N-termini that correspond to the N-termini expected to occur after DPP9 cleavage ["GL-" and "GI-," respectively; (Calvo *et al*, 2017)].

The *in silico* prediction also indicated that substrates of the disulfide relay are overrepresented among the potential mitochondrial DPP8/9 targets. Disulfide relay substrates usually do not contain an N-terminal MTS, and they are imported slowly most likely in a post-translational manner (Fischer *et al*, 2013). This slow import might therefore result in extended dwelling times of fully translated precursors in the cytosol and the risk of premature (mis-)folding (and acquisition of activity) in the wrong compartment. However, these properties would also allow efficient modulation of imported protein amounts by cytosolic N-terminal processing. We indeed observe that for some potential DPP8/9 targets levels change upon DPP8/9 inhibition in HEK293 cells (Fig 7). AK2 levels appear to be

most affected by DPP8/9 inhibition, and this might at least in part also be explained by its slow and complex import and folding pathway that we propose involves intramolecular disulfide isomerization. Isomerization was so far not reported for disulfide relay substrates but frequently occurs during oxidative protein folding in the ER (Jansens *et al*, 2002; Land *et al*, 2003; Chakravarthi & Bulleid, 2004; Roberts *et al*, 2018).

Importantly, DPP8/9-mediated processing might not only influence the stability of its targets. For example, some disulfide relay substrates would expose a glycine after processing that might facilitate myristoylation. Myristoylation has been recently reported to affect mitochondrial import kinetics and multi-subunit complex assembly (Ueda *et al*, 2019).

DPP8/9 processing could also become relevant during pathophysiological conditions. Recent reports indicated that proteasome inhibition might be a promising therapeutic approach for patients that carry protein variants that delay protein import by the disulfide relay (Habich *et al*, 2019a; Mohanraj *et al*, 2019). One of these proteins, NDUFB10 (patient mutation: C107S), which is a part of complex I, exhibited a striking tissue heterogeneity: While the mutant protein was absent from liver and heart, in fibroblasts it was present and functional in mitochondria (Friederich *et al*, 2017). In the present study, we demonstrated that indeed DPP8/9 inhibition led to increased amounts of mitochondrial NDUFB10$^{C107S}$ (Fig 7). DPP8/9 activity was previously shown to be highest in liver, with skin cells harboring only little DPP8/9 activity (Yu *et al*, 2009). Thus, one might speculate that due to efficient DPP8/9 processing, NDUFB10 in liver cells has little chance to become imported when its import mechanism is impaired, as is the case with the C107S variant.

For AK2, we provide evidence that DPP9 provides a novel mode to control dual localization. This might expand to other proteins like, e.g., COX17 for which a dual localization to cytosol and mitochondria has previously been reported (Beers *et al*, 1997; Heaton *et al*, 2001). Compared to another major mode of cytosol-IMS dual localization, competition between folding and import (resulting in high cytosolic and low IMS protein levels), DPP8/9-dependent regulation ensures high IMS protein levels and only a small protein fraction to accumulate in the cytosol. It also allows depleting the cytosol completely of unwanted localized IMS proteins.

To serve in regulation of mitochondrial biogenesis, cellular DPP8/9 activity needs to be tunable. High-throughput studies indicated the potential to regulate DPP9 activity posttranslationally by cysteine residue modification, lysine acetylation, and malonylation or binding of sumoylated proteins (Gorrell *et al*, 2006). One of the modified lysine residues, K43 in DPP9, is situated very close to the substrate-binding site and its modification might well affect substrate binding and link DPP9 activity to the energy state of the cell. Likewise, the conserved cysteine 844 in DPP9 (which is not found in DPP8) is close to the substrate-binding site. Modification of cysteine residues with NEM or hydrogen peroxide inhibited DPP9 activity (Gorrell *et al*, 2006). Thus, it is tempting to assume that specifically DPP9 activity is susceptible to oxidative stress or redox signaling. Lastly, the activity of DPP9 is increased by binding to sumoylated proteins (Pilla *et al*, 2012), integrating the DPP9 processing pathway into other cellular signaling networks. DPP8/9 levels can also be regulated on a transcriptional level. This has been demonstrated, e.g., for the hematopoietic system (Bagger *et al*, 2019) (Appendix Fig S7C). Interestingly, the promoter region of DPP9 contains several confirmed binding sites for heat shock factor 1 (HSF1) (Mendillo *et al*, 2012). HSF1 is the major regulator of the cellular heat shock response but has also been linked to maintaining mitochondrial integrity and metabolic adaptations (Mendillo *et al*, 2012; Santagata *et al*, 2013). Thus, the posttranslational modulation of mitochondrial protein import might well be closely integrated into many cellular processes.

DPP8 and DPP9 have been assigned important roles in different physiological settings including roles in the immune system, in inflammation, in preadipocyte differentiation, and during cancer progression. We demonstrate here an important molecular role for DPP8 and DPP9 in controlling mitochondrial biogenesis. Given the crucial role of mitochondria in many of the above processes, our work will in the future allow detailing the role of DPP8/9-modulated mitochondrial import in understanding physiological outcomes of variations in DPP8/9 activity.

# Materials and Methods

### Plasmids and cell lines

For plasmids and cell lines used in this study, see Appendix Table S1. For the generation of stable, inducible cell lines the HEK293 cell line-based Flp-In T-REx-293 cell line was used with the Flp-In T-REx system (Invitrogen). For siRNA-mediated knockdown experiments and analysis of endogenous protein levels upon 1G244-treatment, HeLa cells were used. Cells were cultured in DMEM supplemented with 8% fetal bovine serum and penicillin/streptomycin at 37°C under 5% $CO_2$.

### Digitonin-fractionation

To assess the localization of AK2 upon inhibition of DPP8/9, 1G244-treated HeLa cells were trypsinized and washed with ice-cold PBS. The PBS was removed, and cells were resuspended in 200 μl fractionation buffer (20 mM HEPES pH 7.4, 250 mM sucrose, 50 mM KCl, 2.5 mM MgCl, 1 mM DTT). Then, the cell suspension was supplemented with 800 μl fractionation buffer containing either 0,

0.003 or 0.03% digitonin and 1× protease inhibitor cocktail (Sigma-Aldrich). To degrade DNA, 25 U Benzonase (Sigma-Aldrich) was added. The samples were incubated for 30 min on ice while being inverted every 5 min. Samples were centrifuged for 10 min at 9,000 *g* at 4°C. Proteins of the supernatant were precipitated using ice-cold TCA. The pellet was resolved in 800 μl fractionation buffer containing 25 μg/ml trypsin, and samples were incubated for 30 min on ice while being inverted every 5 min. Then, proteins of the pellet were precipitated using ice-cold TCA. TCA pellets were dissolved in 80 μl resolving buffer (0.2 M Tris–HCl pH 7.5, 6 M urea, 10 mM EDTA, 2% SDS). Samples were sonicated until pellets were entirely dissolved, supplemented with 4× Laemmli buffer containing 200 mM DTT, and analyzed via SDS–PAGE and immunoblotting.

### Homogenizer fractionation

For separation of cytosol, mitochondria and nuclei, a fractionation protocol based on mechanical shear forces was performed. The whole procedure was carried out at 4°C and with pre-cooled buffers. HEK293-cells were washed with PBS, incubated in hypoosmotic PBS and afterward resuspended in 200 μl STM buffer (250 mM sucrose, 50 mM Tris–HCl pH 7.4, 5 mM $MgCl_2$, protease inhibitor cocktail (Sigma-Aldrich, S8820)). The cells were disrupted in reaction tubes using a Potter–Elvehjem style homogenizer for 1 min. 20% of the lysate were taken as control. Nuclei were separated from the mixture by centrifugation for 3 min at 300 *g*. The nuclear fraction was subsequently washed twice in STM buffer and resuspended in NET-buffer (20 mM HEPES pH 7.9, 5 mM $MgCl_2$, 0.5 M NaCl, 0.2 mM EDTA, 20% glycerol, 1% Triton X-100, protease inhibitor cocktail) and incubated for 30 min. Finally, the nuclear lysate was sonicated to disrupt nuclear DNA. The mixture containing mitochondria and cytosol was further separated by centrifugation at 11,000 *g* for 30 min. The supernatant contains the cytosol and was acetone-precipitated and subsequently resuspended in STM buffer. The mitochondrial sediment was washed twice in STM buffer und finally resuspended in SOL buffer (50 mM Tris–HCl pH 6.8, 1 mM EDTA, 0.5% Triton X-100, protease inhibitor cocktail). Samples were sonicated until pellets were entirely dissolved, supplemented with 4× Laemmli buffer containing 200 mM DTT, and analyzed via SDS–PAGE and immunoblotting.

### Cell treatments

To inhibit the proteasome, cells were cultivated with medium supplemented with MG132. For Western blot analysis, cells were treated with 1 μM MG132 for 16 h, while in pulse-chase experiments treatment with 5 μM MG132 started during the starvation period. To inhibit DPP8/9, cells were supplemented with medium containing 1G244. To assess the effect on HA-tagged proteins, cells were treated with 10 μM 1G244 for 16 h. The effect on endogenous proteins was assessed after incubation of cells with 1 μM 1G244 for 3 or 5 days and renewal of medium each day. Exogenous protein expression using the Flp-In T-REx-293 cell line was induced by incubation of cells for 16 h with medium containing doxycycline (0.01 μg/ml or 1 μg/ml for AK2-variants, 1 μg/ml for all other proteins). For siRNA-mediated knockdown, cells were transfected with siRNA (Qiagen) (Appendix Table S1) using Interferin (Peqlab)

according to the manufacturer's protocol and incubated for 72 h. Emetine chase experiments were performed by addition of 100 μg/ml emetine for the indicated times.

## Steady-state protein levels

To assess steady-state protein levels, cells were washed with ice-cold PBS and lysed either by addition of Laemmli buffer (2% SDS, 60 mM Tris–HCl pH 6.8, 10% glycerol, 0.0025% bromophenol blue) or either LCW buffer (0.5% Triton X-100, 0.5% Na-deoxycholate, 150 mM NaCl, 20 mM Tris pH 7.5, 10 mM EDTA, 30 mM Na-pyrophosphate) or RIPA buffer (50 mM Tris pH 8, 150 mM NaCl, 1% Triton X-100, 0.5% sodium deoxycholate, 0.1% SDS). RIPA and LCW buffers were supplemented with 1× protease inhibitor cocktail (Sigma-Aldrich). Laemmli lysis samples were boiled for 10 min at 96°C and analyzed via SDS–PAGE and immunoblotting. RIPA/LCW lysis samples were incubated for 30 min on ice and then centrifuged for 30 min at 20,000 g at 4°C. The protein concentration of the supernatant was determined via Roti-Quant Universal Kit (Carl Roth). Samples of 50 μg protein were prepared, boiled for 5 min at 96°C, and analyzed via SDS–PAGE and immunoblotting.

## Viability assay

Cell viability was assessed with the PrestoBlue Cell Viability Reagent (Thermo Fisher). To this end, cells were grown in 96-well plates and subjected to different treatments. The PrestoBlue reagent was applied to the cells in a 1:10 dilution and incubated for 1 h at 37°C. Afterward, the fluorescence was measured in a CLARIOstar plate reader with excitation at 560 nm and emission at 595 nm. After the measurement, medium was exchanged to fresh medium without PrestoBlue for continued culture. For analysis, all values were blank-corrected and normalized to their respective untreated control.

## Immunoprecipitation

Immunoprecipitations were carried out either combined with native lysis or with denaturing lysis. For both, cells were washed with PBS containing 20 mM NEM (N-Ethylmaleimide) and afterward incubated at 4°C in PBS-NEM for 10 min. Afterward, the cells were scratched off and sedimented (500 g, 5 min). For native lysis, the cells were resuspended in native lysis buffer (100 mM NaPi pH 8.1; 1% Triton X-100) and lysed on ice for 1 h. Cell debris was sedimented (20,000 g, 4°C, 1 h), and the supernatant was transferred to beads. Denaturing lysis started with resuspension of the cells in denaturing lysis buffer (30 mM Tris pH 8.1, 150 mM NaCl, 1 mM EDTA). For cell lysis, 1.6% SDS was added and the sample boiled at 95°C for 5 min. To lower sample viscosity, 1 μl Benzonase (Sigma-Aldrich, E1014) was added. After lysis, the samples were filled up to 1 ml with denaturing lysis buffer containing 2.5% Triton X-100. The samples were incubated on ice for 1 h followed by centrifugation for 1 h at 20,000 g and 4°C. Supernatants were then transferred to beads. Beads were tumbled with the samples for 16 h at 4°C for both IP variants. After binding, the beads were washed four times with the respective lysis buffer with Triton X-100 and one last time with the respective lysis buffer without Triton X-100. For elution,

washing buffer was removed completely and 1× Laemmli buffer was added, followed by 3 min boiling at 95°C.

## Protein purification

All AK2 variants as well as AK1 were heterologously expressed with a C-terminal 6x-His tag in *Escherichia coli* Rosetta 2 cells. The proteins were purified by Immobilized Metal Affinity Chromatography using Ni-NTA columns (Qiagen), followed by size exclusion chromatography with a 16/600 Superdex 75 pg column (GE). Protein concentration was determined by measuring absorbance at 280 nm and using a calculated extinction coefficient (ProtParam).

## Assay to address inverse and direct redox states of protein thiols

The steady-state redox state assay was performed similarly to what was described before (Erdogan et al, 2018). Reduced thiols were blocked by treatment with 20 mM NEM (N-Ethylmaleimide) dissolved in PBS at 4°C for 15 min for minimal shift (min) and steady-state (ss) samples. Maximum shift (max) samples were treated with PBS at 4°C for 15 min. The cells were washed twice with PBS and finally scratched off in PBS. The cells were sedimented by centrifugation at 500 g for 5 min and lysed in native IP lysis buffer. Immunoprecipitation was performed as described above. After elution of proteins by addition of 1× Laemmli buffer, the proteins were treated with 40 mM TCEP (tris(2-carboxyethyl)phosphine) for 15 min at 65°C before mmPEG12 modification (15 μM final concentration). For min shift controls, mmPEG12 was omitted. After modification, samples were loaded and analyzed on Tris–Tricine–PAGE, Western blotting, and immunodetection.

For determination of progressive oxidation during protein maturation, the *in vivo* oxidation assay was performed. After pulse labeling, the proteins were modified by addition of mmPEG12 (ss samples). The max shift samples were reduced by incubation with 40 mM TCEP for 15 min at 65°C and modification afterward. Min shift samples were left untreated. After modification, the samples were subjected to native immunoprecipitation as described above followed by elution from the beads and analysis on Tris–Tricine–PAGE, Western blotting, and autoradiography.

## Pulse-Chase biogenesis assay

Pulse-chase assays to assess protein oxidation in intact cells (oxidation assay) were performed as described previously (Fischer et al, 2013). Newly synthesized proteins were pulse-labeled with EasyTag EXPRESS $^{35}$S Protein Labeling Mix (Perkin Elmer) at a concentration of 200 μCi/ml. The pulse labeling was stopped by addition of chase medium containing 20 mM methionine and incubation for indicated times. The chase was stopped by adding ice-cold 8% TCA. After TCA precipitation, the pellets were dissolved in resolving buffer (0.2 M Tris pH 7.5, 10 mM EDTA, 6 M Urea, 2% SDS, 0.0025% bromphenol blue) and either used for thiol modification or directly for immunoprecipitation.

## SILAC and mass spectrometry

Stable Isotope Labeling in Cell Culture (SILAC) was performed as described in Petrungaro et al (2015). Cells were subcultured and

passaged in SILAC-DMEM (Thermo Fisher), supplemented with 10% dialyzed FBS (Gibco, Invitrogen), 1% L-glutamine (PAN Biotech), containing either L-arginine or L-arginine-$^{13}C_6$-$^{15}N_2$ (42 mg/l), and L-lysine or L-lysine-$^{13}C_6$ $^{15}N_2$ (73 mg/l), and 27.3 mg/l proline. Cells were lysed by adding native lysis buffer (100 mM NaPi pH 8.0, 100 mM NaCl, 1% Triton X-100). Lysates were incubated for 1 h at 4°C and then centrifuged for 1 h at 25,000 g. The supernatant was subjected to immunoprecipitation with anti-HA beads (Sigma-Aldrich) at 4°C overnight. Samples were washed four times using native lysis buffer and once with lysis buffer without Triton X-100. The collected beads were heated in Laemmli buffer containing 1 mM DTT for 10 min at 95°C. After alkylation using 5.5 mM iodoacetamide for 10 min at room temperature, the samples were centrifuged and the supernatants were loaded on 4–12% gradient gels (NuPAGE, Thermo Fisher) for protein separation. After staining, each gel lane was cut into 5–10 slices, the proteins were in-gel digested with trypsin (Promega) and the resulting peptide mixtures were processed on STAGE tips and analyzed by LC-MS/MS. The LC-MS measurements were performed on a QExactive Plus or HF-X mass spectrometer coupled to an EasyLC 1000/1200 nanoflow-HPLC. Peptides were separated on fused silica HPLC-column tip (I.D. 75 µm, New Objective, self-packed with ReproSil-Pur 120 C18-AQ, 1.9 µm (Dr. Maisch) to a length of 20 cm) using a gradient of A (0.1% formic acid in water) and B (0.1% formic acid in 80% acetonitrile in water): loading of sample with 0% B with a flow rate of 600 nl/min; separation ramp from 5 to 30% B within 85 min with a flow rate of 250 nl/min). For nanoESI, the spray voltage was set to 2.3 kV and ion-transfer tube temperature to 250°C, no sheath and auxiliary gas was used. The mass spectrometer was operated in the data-dependent mode; after each MS scan (mass range $m/z$ = 370–1,750; resolution: 70,000 for Plus and 120,000 for HF-X) a maximum of ten and 12 MS/MS scans were performed using a normalized collision energy of 25%, a target value of 1,000 and 5,000 and a resolution of 17,500 and 30,000, for the Plus and HF-X mass spectrometers, respectively. The MS raw files were analyzed using MaxQuant Software version 1.4.1.2 (Cox & Mann, 2008) for peak detection, quantification, and peptide identification using a full-length UniProt human database (March, 2016) and common contaminants such as keratins and enzymes used for in-gel digestion as reference. Carbamidomethylcysteine was set as fixed modification and protein amino-terminal acetylation and oxidation of methionine were set as variable modifications. The MS/MS tolerance was set to 20 ppm, and three missed cleavages were allowed using trypsin/P as enzyme specificity. Peptide and protein FDR based on a forward-reverse database were set to 0.01, minimum peptide length was set to 7, and minimum number of peptides for identification of proteins was set to one, which must be unique. The "match-between-run" option was used with a time window of 1 min.

## In vitro AK2 activity assays

For the forward activity assay, AK activity was coupled to pyruvate kinase (PK) and lactate dehydrogenase (LDH, PK/LDH mix from Sigma-Aldrich). In this coupled reaction, ADP produced by AKs is used by PK to convert phosphoenolpyruvate to pyruvate, which is used by LDH to oxidize NADH, thus decreasing the absorbance at 340 nm. The following reaction conditions were used: 50 mM Tris/ HCl pH 7.5, 50 mM KCl, 4 mM MgCl₂, 0.006% BSA, 0.25 mM NADH, 1 mM PEP, 1 mM ATP. The concentration of AK2 was set to 0.8 nM, AK1 to 0.4 nM.

The reverse activity assay was carried out similarly to the forward activity assay, by coupling the AK reaction to hexokinase (HK) and glucose-6-phosphate dehydrogenase (G6PDH, HK/G6PDH mix from Roche). In this assay, AKs provide the ATP for glucose phosphorylation by HK, followed by NADP$^+$ reduction to NADPH and an increase in absorbance at 340 nm. The reaction conditions were as follows: 58 mM glycylglycine pH 7.4, 10 mM MgCl₂, 0.006% BSA, 0.25 mM NADP$^+$, 20 mM glucose. The concentration of AK2 was set to 4 nM, AK1 to 0.8 nM. All measurements were performed in triplicates in 96-well plates and read in a CLARIOstar microplate reader set to 25°C. A measurement without AK was performed simultaneously to all measurements to allow subtraction of the background reaction.

## Cellular AK2 activity assay

AK2 activity in HEK293 cells was assessed by performing the aforementioned coupled enzyme assay with immunoprecipitated AK2-HA. HEK293 cells overexpressing AK2-HA variants were lysed in native lysis buffer (100 mM NaPi pH 8.1, 0.2% n-Dodecyl-D-maltoside) for 20 min on ice. AK2-HA was purified from the supernatants by incubation for 2 h with anti-HA agarose beads (Sigma). After washing, the beads were resuspended in ADP assay buffer containing 0.38 mM NADP$^+$ and 1 mM ADP. The reaction mix was then incubated and shaken at 25°C. For each timepoint, some supernatant was taken after a short centrifugation and transferred to 96-well plates for absorbance measurement. Additionally, a reaction mix without added beads was used as negative control.

## Immunofluorescence and colocalization analysis

HEK293 Flp-In T-REx cells expressing HA-tagged AK2 mutants were cultured on poly-L-lysine-coated cover slips for 48 h. Protein expression was induced by addition of 1 µg/ml doxycycline after 24 h of culture. Before fixation, the cells were incubated with MitoTracker Red (Thermo Fisher) for 1 h at 37°C. Fixation was performed with 4% paraformaldehyde for 15 min. Then, cell membranes were permeabilized with blocking buffer (20 mM HEPES pH 7.4, 3% BSA, 0.3% Triton X-100) for 1 h. Cells were washed and incubated with primary (anti-HA, 3F10, Roche) and secondary antibodies (anti-rat, AlexaFluor 488) for 16 h at 4°C and 1 h at room temperature, respectively. Then, the cells were washed and the cover slips transferred to microscope slides and mounted in ProLong Gold Antifade mountant containing DAPI. Afterward, the cells were analyzed by confocal fluorescence microscopy. The images were analyzed for correlation of the HA signals and the MitoTracker signals with CellProfiler (Carpenter et al, 2006). In the analysis, cellular outlines are defined by thresholding and watershedding followed by measurement of the Pearson's correlation coefficient in the defined outlines. The results of the pipeline were checked manually for suitability afterward (e.g., clumped cells which are indistinguishable were excluded). Representative pictures were adjusted for the respective intensity ranges with Fiji (Schindelin et al, 2012). Statistical analysis was performed in RStudio using the Kruskal–Wallis

rank sum test followed by Dunn's test adjusted with the Benjamini–Hochberg method.

### Thermofluor assay

The thermal stability of proteins was determined by using the thermal shift fluorescence assay (Niesen *et al*, 2007). The assay was performed in a CFX 96 Real-Time PCR Detection Systems (Bio-Rad Laboratories). Sypro Orange (S5692, Sigma-Aldrich) was used in a 1:5,000 dilution. Melting curves were obtained while ramping the temperature from 20 to 95°C at a rate of 1°C/min in steps of 0.5°C every 30 s. Fluorescence was monitored using "FRET" scan mode and was measured after each step. The melting temperature (Tm) of a protein is defined as the inflection point of the unfolding transition; or in other words, as the minimum of the negative first derivative of the melting curve. Tm was determined by using the internal algorithms of the data analysis software (Bio-Rad CFX Manager™, Version: 3.1.1517.0823).

### Conservation and logo plots

Amino acid conservation analysis was performed by using protein sequences provided by UniProt and aligning them using Jalview (Waterhouse *et al*, 2009). Alignment of sequences was achieved with help of the algorithm of MUSCLE (Edgar, 2004). For the AK2 N-terminus comparison, sequences from 163 species were aligned, while for the other proteins 11 species each were compared. The conservation is depicted as logo plot, where the relative sizes of the letters correspond to the relative abundance of the respective residue at the given position. Additionally, the consensus sequence is shown below the logo plots.

### Quantification and statistical analysis

Intensity of autoradiography and immunoblot signals were quantified using ImageQuantTL (*GE*) and Image Laboratory (Bio-Rad Laboratories), respectively. For SILAC MS data interpretation, Perseus software was used (Tyanova *et al*, 2016). Correlation analysis was performed with CellProfiler (Carpenter *et al*, 2006) and statistical analysis of correlation analyses with RStudio. Analysis of amino acid conservation was performed with Jalview. Error bars in figures represent standard deviation. The number of experiments is reported in the figure legend.

## Data availability

The authors declare that there are no primary datasets and computer codes associated with this study.

**Expanded View** for this article is available online.

### Acknowledgements

We thank Laura Herold for initial help with cell-based AK2 assay and Ruth Geiss Friedländer for help in the initial stages of the project. We thank Anja Wittmann for technical help. The Deutsche Forschungsgemeinschaft (DFG) funds research in the Laboratory of JR (RI2150/2-2—project number 251546152, RI2150/5-1—project number 435235019, CRC1218/TP B02—project number 269925409, and RTG2550/1—project number 411422114), DM (TRR130/TP03), and BB (FOR2722/TP1/BB—project number 384170921, BR2304/12-1—project number 407146744). MH was a PhD fellow of the Carl-Zeiss Stiftung. Carmelina Petrungaro was a PhD fellow of the Boehringer Ingelheim Fonds. We also thank the CECAD imaging facility. Open access funding enabled and organized by Projekt DEAL.

### Author contributions

JR, MH, and YF designed the study and planned experiments. MH, YF, SG, JK, LS, KJL, EvdL, MA, CP, SLS, CP, and SU performed experiments and analyzed data. KH performed *in silico* analysis. JD performed proteomic analysis. JR, MH, and YF designed figures. JR, MH, and YF wrote the manuscript with critical input from UB, BB, KH, and DM.

### Conflict of interest

The authors declare that they have no conflict of interest.

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
