## [Review Process File · The EMBO Journal]

Proteasomal degradation induced by DPP9-mediated processing competes with mitochondrial protein import

Yannik Finger, Markus Habich, Sarah Gerlich, Sophia Urbanczyk, Erik van de Logt, Julian Koch, Laura Schu, Kim Lapacz, Muna Ali, Carmelina Petrunaro, Silja Salscheider, Christian Pichlo, Ulrich Baumann, Dirk Mielenz, Jörn Dengjel, Bent Brachvogel, Kay Hofmann, and Jan Riemer
DOI: [10.15252/emj.2019103889](https://doi.org/10.15252/emj.2019103889)

Review Timeline:

Submission Date:	1st Nov 19
Editorial Decision:	17th Dec 19
Revision Received:	30th May 20
Editorial Decision:	24th Jun 20
Revision Received:	29th Jun 20
Accepted:	3rd Jul 20

Editor: Elisabetta Argenzio

Transaction Report:

Thank you for submitting your manuscript entitled "Cytosolic processing of mitochondrial precursors by DPP9 controls mitochondrial protein import" [EMBOJ-2019-103889] to The EMBO Journal. Please accept my apologies for the delay in communicating our decision. Three referees were originally assigned to your manuscript, however one of them did not return his/her report even after repeated chasing messages. The two referee reports that we have received are enclosed below for your information.

As you can see, the referees find your study interesting and raise a few points that have to be addressed before they can support the publication of your work in The EMBO Journal. In particular, referee #1 requests you to properly quantify all experiments and to clarify the inconsistencies present in the study. Reviewer #3 asks you to test the interaction between endogenous CHCHD4 and AK2 and to show that unrelated IMS proteins are not pulled down by CHCHD4 and AK2. This reviewer also demands you to verify whether cytosolic accumulation of AK2 is harmful to the cell.

Given the overall interest of your study, I would like to invite you to revise the manuscript in response to the referee reports. I should note that conclusively addressing these and all the other referees' points is essential for publication in The EMBO Journal.

When preparing your letter of response to the referees' comments, bear in mind that this will form part of the Review Process File and will be available online to the community. For more details on our Transparent Editorial Process, please visit our website:
http://emboj.embopress.org/about#Transparent_Process.

We generally grant three months as standard revision time. As a matter of policy, competing manuscripts published during this period will not negatively impact on our assessment of the conceptual advance presented by your study. Nevertheless, please contact me as soon as possible upon publication of any related work.

Referee #1:

The paper presents the discovery of a novel mechanism in the field of mitochondrial protein import. A major finding here is that a cytosolic protease can regulate the biogenesis of mitochondrial proteins, particularly proteins located in the IMS which follow the cysteine oxidation pathway by CHCHD4 (human form of Mia40). As exemplified with the adenylate kinase AK2 as substrate protein its processing as precursor protein by the dipeptidylpeptidases DPP9/8 in the cytosol is crucial for proteasomal degradation when they are too long in the cytosol (conditions where their import into mitochondria is impaired) following the N-end rule of the cytosol. The mechanism

presented here can be considered as a novel concept in mitochondrial protein biogenesis as it demonstrates for the first time the involvement of cytosolic proteolytic processing of mitochondrial precursor proteins in regulation of their import (so far processing of precursors were only known to happen inside the organelle). The experimental data are of very high quality and the manuscript is well written (although the text could be smoothen a bit to make the reading/understanding a bit easier for non-specialists). The paper fits very well for EMBO journal. However there are a few points that would be helpful to clarify prior publication.

- some graphics of quantifications contain statistical analysis but others not. For example: Fig3B the statistical analysis is missing for the control vs MG132 experiment. The same with Fig3D. Please comment or add the data if available.

- Fig 4D: Depletion of DPP9 levels produces an increase also in the levels of Wt AK2. How is that possible if in principle the DPP9 processing occurs in the cytosol and the wt AK2 is import competent and only located in mitochondria (Fig2)? Moreover, in the strain that has CHCHD4 depleted one would assume that upon conditions where AK2 import is impaired and its localization shifts to the cytosol. Then when the protease DPP9 is depleted the stabilization and increase of AK2 levels should be higher than in the control. Even one could say the AK2 levels don't look higher than control in the cell line with DPP9 and CHCHD4 siRNA, however the quantification bar in the graphic is higher. Could the authors explain this?

Very similarly this happens upon proteasome inhibition (Fig3D). Here the authors show an increase in the AK2 levels in western blot but in the immunofluorescence data in Fig S5 C and I they state "cytosolic localization of AK2 C40S but not AK2 wildtype increases upon proteasomal inhibition". If we see an increase of the levels by western blot, and the MG132 treatment should increase cytosolic protein levels only (because they are not degraded by the proteasome in the cytosol), then why don't we see an increase of the cytosolic fraction of AK2 wt upon MG132 treatment using immunofluorescence?

Did the authors also test the cell fractionation assay of Fig. 5A upon MG132 treatment? Would they see then AK2 accumulation? This would be a nice complement (but not really necessary).

In Figs. 6C and D the WT variant is missing in the graphics in order to claim that the cysteine mutants don't have any influence in AK2 activity.

In addition in the cellular AK2 activity assay in Fig. 6E, input/loading controls would be good to show. To check if the activity is affected or not in the AK2 variants, a loading control ensuring that the same amount of protein is added to the assay would be good. One could assume that the cells expressing the different variants of AK2 have different levels of the protein as observed in previous figures. It needs to be clarified that the observed differences in the assay are due to differences in the AK2 protein levels or to the activity itself.

Fig.7 D and E: the authors claim that the NDUFB10 C107S variant with an impaired import capacity is stabilized upon 1G244 (DPPs inhibitor). Indeed it is but no more stabilized than the Wt version that is imported without delay. So the conclusion that a version that imports with less efficiency is affected and stabilized by inhibiting DPPs proteases is disputatious because the WT version is affected in the same way. Therefore the conclusions commented during the discussion about NDUFB10 C107S version stabilization upon DPPs inhibition is should be phrased a bit more careful.

The experiments are performed in cells growing in DMEM (media with glucose). Would the redox

status of AK2 cysteines in cells grown in galactose be different (conditions that promote mitochondria biogenesis? Would the accumulation of AK2 upon DPP9 depletion under such conditions be different? A comment on this would be helpful.

Did the authors check if AK2 precursor might also contain additional internal mitochondrial targeting signals (iMITS)? Like shown in Backes et al, 2018.

Fig. S2 E signal of the 60min time point for AK2 lysate is decreasing dramatically during the oxidation kinetic? Can the authors explain why?

Could such a mechanism as the authors discovered here also exist in yeast, i.e. did they checked potential Dipetidylpeptidase derived processing intermediates in the N-proteome data which exist for yeast and are DPP8/9 cleavage site motifs also present in some IMS proteins from yeast?

Referee #3:

Members of the dipeptidyl peptidase (DPP) family are N-terminal dipeptide serine proteases. Recent findings show a role for DPP8/9 in the immune system and in preadipocyte differentiation. In addition, these peptidases were suggested to be essential for neonatal survival and to play a role in antigen maturation, cell migration, and cell adhesion. DPP9 activity was also connected to pathophysiological conditions by promoting tumorigenicity and metastasis in nonsmall cell lung cancer. In their contribution, the authors suggest yet another function for these peptidases namely, in the regulation of the mitochondrial import of a subset of mitochondrial proteins. They show first that AK2 is imported into the mitochondrial IMS by the MIA40 pathway. In addition, the authors demonstrate that DPP8/9 is processing AK2 in the cytosol and that the newly formed N-terminal of Ak2 directs the protein to degradation by the proteasome. They finally suggest that such a mechanism can help to regulate protein import into mitochondria. The experimental work is generally of high quality and these findings are of interest. However, some issues should be addressed.

Major points:

- 1) Figs. 1B and S1B: (i) I guess that both CHCHD4 and HA-AK2 were overexpressed in these cell lines. It will be helpful if the authors show interaction with endogenous levels of AK2. In addition, the interaction between the two proteins takes probably place in the mitochondrial IMS. Thus, the authors should demonstrate that an unrelated IMS protein would NOT be pulled-down by either CHCHD4-strep or HA-AK2.
(ii) It seems that native CHCHD4-Strep was expressed and bound to the beads in higher amounts than the C53S/C55S variant. Can the authors exclude the possibility that this difference explains why a binding of AK2 to the mutated variant was not detected?
- 2) Fig. 2: AK2 was suggested previously to have dual-location in the mitochondrial IMS and cytosol. Does the observation that all molecules of AK2 are oxidized in cysteine C42 and C92 suggest that ALL AK2 molecules in the cell passed through the mitochondrial IMS or are actually there? Is there under normal conditions any cytosolic functional population of AK2?
- 3) Figs. 3 and 4: The inhibition of the proteasome stabilizes the cysteine AK2 mutant by 45 fold (Fig. 3D) whereas inhibiting DPP8/9 stabilizes it by 14 fold (Fig. 4D). How do the authors explain this difference in light of their claim that DPP8/9 processing of AK2 is required for proteasomal degradation?
- 4) The authors nicely show that DPP8/9 can process AK2 to reduce its accumulation in the cytosol.

However, the authors do not explain why such cytosolic accumulation can be harmful. It will improve the MS if the authors will indeed test whether such accumulation is harmful.

5) Fig. 4G: the effect of 1G244 in this experiment is not convincing. At Day #2 there is no difference upon addition of the reagent and also in Day #1 the only effect is seen in lane #10. Results that are more convincing should be presented or this part should be omitted.

6) The title is not supported by the actual findings. The authors show neither that the processing by DPP8 is regulated nor that it controls protein import. At the most one can claim that processing competes import. Thus, I suggest to modify the title.

Minor points:

(a) Figs. 1B and S1B: Was the immunodetection of HA-AK2 done with antibodies against AK2 or against the HA-tag? If the latter, these should be indicated in the figure. In all IP experiments it should be added to the corresponding legend which percentage of the lysate and eluate were loaded on the gel.

(b) Fig. S1: Using siRNA against AK2-A resulted in a weaker upper band (middle panel). However, this reduction cannot be described as "the upper band disappears". In the right panel the resolution of lane #1 is not good enough to know which, if any, band disappears. Thus, the text should be modified.

(c) Fig. 1E: It will be helpful to show a Western of CHCHD4 to estimate the efficiency of the knockdown. In addition, it seems that the oxidation of isoform A is not affected by depletion of CHCHD4. Can the authors explain this observation?

(d) Page 7: According to the results of Fig. 2 that show rather minor contribution of C40, it is misleading to describe at this stage the influence of C40 as "critical".

(e) Fig. S2D: To evaluate specificity in the binding, the intensities of the bands of CHCHD4 and CPOX (at least in the input lanes) should be similar. Hence another panel should be presented. Does the observation that the input amounts of WT and C40S are similar suggest that interaction with CHCHD4 does not contribute to the overall stability of AK2? The authors should discuss this point.

(f) Fig. 3A: lanes 2-3 and 4-5 are labeled the same. Are these two different cell lines for each treatment? This point should be explained in the legend.

(g) Fig. 3D: It seems that WT AK2 is also stabilized upon addition of MG132. On the other side, also in the presence of this proteasome inhibitor the levels of the cysteine mutant is lower than that of the WT protein. Thus, are other proteases (in addition to the proteasome) involved in the removal of the mutant protein? The authors should elaborate on this point.

(h) Fig. 7D, E: The contribution of 1G244 to the total and mitochondrial amounts of NDUFB10-C107S is very similar. Are there in total more mitochondria in 1G244-treated cells?

(i) Page 14, 1st paragraph: Table S1 is mentioned here but this table does not contain information on potential DPP8/9 substrates.

(j) A list in the Supplementary Information part describing the source of the used antibodies will be helpful.

Point-by-Point Response, Habich, Finger et al

As an additional notion: The nomenclature for the protein **CHCHD4/MIA40** is somewhat inconsistent in mammalian cells. We thus decided to stick to the name of the protein that is most commonly used and also describes its function best, MIA40. Thus, throughout the manuscript, we now renamed CHCHD4 into **MIA40**.

Referee #1:

*The paper presents the discovery of a novel mechanism in the field of mitochondrial protein import. A major finding here is that **a cytosolic protease can regulate the biogenesis of mitochondrial proteins**, particularly proteins located in the IMS which follow the cysteine oxidation pathway by CHCHD4 (human form of Mia40). As exemplified with the adenylate kinase AK2 as substrate protein its processing as precursor protein by the dipeptidylpeptidases DPP9/8 in the cytosol is crucial for proteasomal degradation when they are too long in the cytosol (conditions where their import into mitochondria is impaired) following the N-end rule of the cytosol. The mechanism presented here can be considered as a novel concept in mitochondrial protein biogenesis **as it demonstrates for the first time the involvement of cytosolic proteolytic processing of mitochondrial precursor proteins in regulation of their import** (so far processing of precursors were only known to happen inside the organelle). The experimental data are of very high quality and the manuscript is well written (although the text could be smoothen a bit to make the reading/understanding a bit easier for non-specialists). The paper fits very well for EMBO journal. However, there are a few points that would be helpful to clarify prior publication.*

We thank this referee for appreciating the novelty and quality of our work.

- some graphics of quantifications contain statistical analysis but others not. For example: Fig3B the statistical analysis is missing for the control vs MG132 experiment. The same with Fig3D. Please comment or add the data if available.

In the previous version of the manuscript, we had in some instances too few biological replicates to perform proper statistical analyses. For the revised manuscript, we have performed additional experiments to increase the number of biological replicates and carefully quantified all experiments. Moreover, we provide statistical analysis for these experiments. This led to changes/improvements in **Figures 3A, 3C, 3D, 3E, 4B, 4C, 4D, 4E, 4F, 5A, 7D, S2D, S2E, S3A, and S4D**. Additional experiments (biological replicates) were thereby performed for **Figures 3A, 3C, 3E, 4E, S2E, and S5B**. The expansion of data and the statistical analysis confirmed all our findings and statements from the previous version of the manuscript.

- Fig 4D: Depletion of DPP9 levels produces an increase also in the levels of Wt AK2. How is that possible if in principle the DPP9 processing occurs in the cytosol and the wt AK2 is import competent and only located in mitochondria (Fig2)? Moreover, in the strain that has CHCHD4 depleted one would assume that upon conditions where AK2 import is impaired and its localization shifts to the cytosol. Then when the protease DPP9 is depleted the stabilization and increase of AK2 levels should be higher than in the control. Even one could say the AK2 levels don't look higher than control in the cell line with DPP9 and CHCHD4 siRNA, however the quantification bar in the graphic is higher. Could the authors explain this?

Very similarly this happens upon proteasome inhibition (Fig3D). Here the authors show an increase in the AK2 levels in western blot but in the immunofluorescence data in Fig S5 C and I they state "cytosolic localization of AK2 C40S but not AK2 wildtype increases upon proteasomal inhibition". If we see an increase of the levels by western blot, and the MG132 treatment should increase cytosolic protein levels only (because they are not degraded by

the proteasome in the cytosol), then why don't we see an increase of the cytosolic fraction of AK2 wt upon MG132 treatment using immunofluorescence?

These are very good points. DPP9 processing does affect wildtype AK2. Like many MIA40 substrates, AK2 is imported posttranslationally and thus exists as precursor for a certain time in the cytosol. During this time of transit to mitochondria, it is not only a target for DPP9 processing (as indicated by our findings that wildtype AK2 contains the processed N-terminus, **Figure 4A**) but also for proteasomal degradation. Thus, inhibition of DPP9 also prevents processing of wildtype AK2 and its degradation in the cytosol. This in turn results then in higher amounts of remaining AK2 precursor that can become imported and, in the end, to increased amounts of mature IMS-localized AK2.

How does MIA40 depletion affect substrate import? MIA40 depletion leads to strongly impaired mitochondrial import of its substrates (Fischer et al, MBoC 2013, Hangen et al, Mol Cell 2015 and Habich et al, Cell Reports 2019). MIA40 serves as *trans-site* import receptor and oxidoreductase for its substrates and once missing, substrates cannot enter mitochondria and become degraded in the cytosol. This leads to strongly diminished steady state levels of these substrates (**Figures 3A** and **4D**). DPP9 inhibition (siRNA depletion) leads to a stabilization of the protein (**Figure 4D**, compare lanes 3 and 4). Inhibition of DPP9 can however not overcome the loss of MIA40 (i.e. lead to import of AK2 into mitochondria). Thus, AK2 remains unfolded in the cytosol and becomes eventually degraded. This is different for the control situation: here, AK2 is stabilized in the cytosol and can become imported into the IMS, where it is matured and stable, leading to a higher increase in AK2 amounts. This is the reason, why we think the combination siMIA40/siDPP9 does not lead to significantly increased levels compared to the control (**Figure 4D**, compare lanes 1 and 4).

Lastly, we agree with the referee in that we also expected the stabilization of AK2^{WT} to lead to a partial accumulation of the protein in the cytosol. At least with our immunofluorescence assays that allow a very thorough statistical analysis due to the high n-numbers this is not the case (**Figure 5C**), although a slight tendency can be observed (the assay might per se also be less sensitive as cytosolic proteins disperse throughout the cytosol and thus signal is lost in the background). In contrast, the A2D and S4P variants of AK2^{WT}, which also become stabilized, do partially relocalize to the cytosol (**Figure 5B**).

Did the authors also test the cell fractionation assay of Fig. 5A upon MG132 treatment? Would they see then AK2 accumulation? This would be a nice complement (but not really necessary).

We would have loved to perform this experiment. However, when we tried to fractionate cells after extended times of MG132 treatment, fractionation led to unreproducible results, *i.e.* the mitochondrial OMM regularly opened quickly and this contaminated the cytosolic fraction. Since we already provide orthogonal approaches to demonstrate the accumulation of a small fraction of cytosolic AK2 upon preventing DPP9 processing or cytosolic degradation, we decided to not further pursue this line of experiments.

In Figs. 6C and D the WT variant is missing in the graphics in order to claim that the cysteine mutants don't have any influence in AK2 activity.

We now included the wildtype data as a new supplemental Figure (new **Figures S6D,E**) and we also explain our reasoning why we used the indicate variants of AK2 for activity analysis.

Expression and purification of the AK2 wildtype from *E. coli* yielded pure, stable and monomeric protein. However, when we purified cysteine variants of AK2, they tended to form disulfide-linked dimers that we never observed in cells (new **Figure S6D**). These dimers

depended on the presence of C232. When we mutated C232, we also lost the dimer and only obtained monomeric AK2 variants.

Thus, we faced the dilemma that if we wanted to analyse the importance of the structural disulfide bond for AK2 for activity, we would have to compare monomeric wild type AK2 with the dimeric AK2^{C42S,C92S} variant. To solve this dilemma and to avoid artefacts from AK2 dimerization, we decided to perform experiments with mutants lacking either all cysteines (i.e. no structural disulfide bond) or lacking all cysteines except for the two cysteines forming the structural disulfide bond.

As an initial control, we compared the activities of AK2^{WT} and AK2^{C40S,C232S}. We found them to be very similar (new **Figure S6E**) indicating that removing C40 and C232 did not impact on activity and thereby validating our approach.

In addition, in the cellular AK2 activity assay in Fig. 6E, input/loading controls would be good to show. To check if the activity is affected or not in the AK2 variants, a loading control ensuring that the same amount of protein is added to the assay would be good. One could assume that the cells expressing the different variants of AK2 have different levels of the protein as observed in previous figures. It needs to be clarified that the observed differences in the assay are due to differences in the AK2 protein levels or to the activity itself.

This is a very good point. With our data with purified proteins from *E. coli*, we already demonstrated that the activity profiles of AK2 with and without cysteines are very similar. Thus, we would expect that any activity differences observed with the assay testing “cell-isolated” AK2 are due to differences in level.

We now included as new **Figure 6F**, data on the relative amounts of AK2 that we used in the assay in **Figure 6E**. This assay was performed the following: HEK293 cells stably expressing the indicated AK2-HA variants were lysed with a DDM-containing buffer and AK2 was precipitated using HA-antibody beads. After the IP these beads were washed and the beads with coupled AK2 were used in the assay shown in **6E**. After the assay, we boiled AK2 off the beads and analysed this lysate by immunoblot against AK2 as shown in **Figure 6F**. Of course, the full performance of the AK2 activity assay on these beads might well lead to the loss of some AK2 from the beads, and the resulting specific activities thus have to be interpreted with care. When we normalized the AK2 activity for the levels of proteins, we found that specific AK2 activity was relatively similar in all cases (bar diagram in **Figure 6F**).

Figure 6E and F thus allow the following statements: 1/ cytosolic AK2 is active, 2/ cytosolic AK2 has likely the same specific activity as IMS AK2., 3/ inhibition of DPP9 increases the levels of AK2 but not the specific activity of the enzyme.

Fig.7 D and E: the authors claim that the NDUFB10 C107S variant with an impaired import capacity is stabilized upon 1G244 (DPPs inhibitor). Indeed it is but no more stabilized than the Wt version that is imported without delay. So the conclusion that a version that imports with less efficiency is affected and stabilized by inhibiting DPPs proteases is disputatious because the WT version is affected in the same way. Therefore the conclusions commented during the discussion about NDUFB10 C107S version stabilization upon DPPs inhibition is should be phrased a bit more careful.

The referee is correct, and we rephrased the text. We wanted to emphasize that protein levels of this NDUFB10 version, which has difficulties to become imported into mitochondria did also increase in mitochondria. This happens to the same extent for the wildtype, too. We have now emphasized this latter point in the text. We also adjusted the quantification of this experiment,

and all sample are now normalized to the DMSO-treated wildtype thus reflecting better the relative amounts of NDUFB10 variants among each other.

The experiments are performed in cells growing in DMEM (media with glucose). Would the redox status of AK2 cysteines in cells grown in galactose be different (conditions that promote mitochondria biogenesis)? Would the accumulation of AK2 upon DPP9 depletion under such conditions be different? A comment on this would be helpful.

We performed both experiments and show them here for the referees. On glucose and on galactose, AK2 is semioxidized, i.e. the C42-C92 disulfide is formed. There might be a small shift in the abundance of the AK2 isoforms (ration of the bands indicated with A and B). We did however not follow up on this.

Moreover, changes of levels of AK2 upon inhibition of DPP9 were similar between glucose- and galactose-grown cells.

Did the authors check if AK2 precursor might also contain additional internal mitochondrial targeting signals (iMTS)? Like shown in Backes et al, 2018.

We have indeed checked this, and yes AK2 contains such a signal. We systematically analysed all human mitochondrial proteins for such signals and are working on this part in a separate project. Thus, we would like to avoid to add this information to this manuscript.

Fig. S2 E signal of the 60min time point for AK2 lysate is decreasing dramatically during the oxidation kinetic? Can the authors explain why?

The experiment depicted in **Figure S2E** has been performed with AK2-HA variants stably expressed in HEK293 cells. The dramatic decrease in levels that is even more pronounced for the C40S variant is due to proteasomal degradation of AK2. For tagged AK2 variants, the effects of proteasomal inhibition were thereby always stronger indicating that imbalanced levels of AK2 are tightly controlled. The dramatic decrease in levels that is even more pronounced for the C40S variant. We repeated the experiment shown in **Figure S2E** multiple times in different settings (*i.e.* different induction times, pulse times, etc.) and always observed this behavior. Importantly, the notion that C40S is more affected by degradation than the wildtype and that the ratio of oxidized to reduced AK2 is tilted towards the reduced form of AK2^{C40S} support a role of C40 in improving import and oxidation efficiency of AK2.

Could such a mechanism as the authors discovered here also exist in yeast, i.e. did they checked potential Dipetidylpeptidase derived processing intermediates in the N-proteome data

which exist for yeast and are DPP8/9 cleavage site motifs also present in some IMS proteins from yeast?

DPP9 is not conserved to yeast. There are according to the SGD two enzymes exhibiting dipeptidyl peptidase activity in yeast, DAP2 and YOL057W. We also checked the mitochondrial N-terminome presented in Voegtle et al, Cell 2009. We could not identify processing events after the 3 or 4 most N-terminal amino acids that occurred in a consensus motif.

Referee #3:

*Members of the dipeptidyl peptidase (DPP) family are N-terminal dipeptide serine proteases. Recent findings show a role for DPP8/9 in the immune system and in preadipocyte differentiation. In addition, these peptidases were suggested to be essential for neonatal survival and to play a role in antigen maturation, cell migration, and cell adhesion. DPP9 activity was also connected to pathophysiological conditions by promoting tumorigenicity and metastasis in nonsmall cell lung cancer. In their contribution, the authors suggest yet another function for these peptidases namely, **in the regulation of the mitochondrial import of a subset of mitochondrial proteins**. They show first that AK2 is imported into the mitochondrial IMS by the MIA40 pathway. In addition, the authors demonstrate that DPP8/9 is processing AK2 in the cytosol and that the newly formed N-terminal of AK2 directs the protein to degradation by the proteasome. They **finally suggest that such a mechanism can help to regulate protein import into mitochondria**. The experimental work is generally of high quality and these findings are of interest. However, some issues should be addressed.*

We also thank this referee for appreciating the novelty and quality of our work.

Major points:

1) *Figs. 1B and S1B: (i) I guess that both CHCHD4 and HA-AK2 were overexpressed in these cell lines. It will be helpful if the authors show interaction with endogenous levels of AK2. In addition, the interaction between the two proteins takes probably place in the mitochondrial IMS. Thus, the authors should demonstrate that an unrelated IMS protein would NOT be pulled-down by either CHCHD4-strep or HA-AK2.*

(ii) It seems that native CHCHD4-Strep was expressed and bound to the beads in higher amounts than the C53S/C55S variant. Can the authors exclude the possibility that this difference explains why a binding of AK2 to the mutated variant was not detected?

We found AK2 in the interactome of MIA40 (Petrungaro et al, Cell Metab 2015). We confirmed this interaction by multiple approaches always involving one of the partners being tagged (**Figures 1B, S1B, S1E**). We now also include an experiment in which we probe for the interaction of the endogenous proteins (new **Figure S1C**). We did this experiment in the form of a radioactive pulse experiment coupled to an IP-reIP approach because the share of AK2 interacting with MIA40 at any time is comparatively low. This is due to the fact that MIA40 helps AK2 to become imported and folded and then release the protein again (see **Figure S1E**). When we precipitated endogenous MIA40 from HEK293 cells after pulse-labelling and then re-immunoprecipitated endogenous AK2 from this precipitate, we could indeed recover both MIA40 and AK2. Both proteins are linked by a disulphide bond (which is opened during reducing SDS-PAGE) and therefore form a covalently linked complex during AK2 maturation.

We also followed the referees advise and redecorated some of our immunoblots against IMS proteins that should not interact with AK2 or MIA40. We now show in new panels in **Figures S1B** and **S1E** that the IMS proteins CPOX and SMAC do not coprecipitate with AK2 and MIA40, respectively.

We think it is very unlikely that MIA40^{C53S,C55S} interacts with AK2 as the interaction between both proteins has to proceed via a disulfide bond, and the experiment shown in **1B** is a denaturing IP that only leaves these covalent interactions intact. With the redox-active cysteines in MIA40 mutated, there is no possibility for this MIA40 variant to interact with AK2 covalently. There is essentially no signal in lane 4 of **Figure 1B** (*i.e.* we cannot even normalize the precipitated AK2 to the precipitated MIA40 levels).

2) *Fig. 2: AK2 was suggested previously to have dual-location in the mitochondrial IMS and cytosol. Does the observation that all molecules of AK2 are oxidized in cysteine C42 and C92 suggest that ALL AK2 molecules in the cell passed through the mitochondrial IMS or are actually there? Is there under normal conditions any cytosolic functional population of AK2?*

We only observe wild type AK2 under normal unperturbed conditions in the IMS, at least in glucose-cultured HEK293 cells. The localization of AK2 might become during differentiation or in different tissues indeed cytosolic, if as we hypothesized in the discussion the activity of DPP9 is attenuated and folding takes place before mitochondrial import can occur. In principle, AK2 that folds in the cytosol does not contain a disulfide bond (the electrons for the oxidation are normally transferred to the oxidoreductase MIA40 in the IMS). Folding does also not require the presence of a disulfide bond as we show with our AK2 activity assays. We can however not exclude that with time a disulfide bond is acquired in AK2 even in the cytosol, e.g. through ROS-induced oxidation.

3) *Figs. 3 and 4: The inhibition of the proteasome stabilizes the cysteine AK2 mutant by 45 fold (Fig. 3D) whereas inhibiting DDP8/9 stabilizes it by 14 fold (Fig. 4D). How do the authors explain this difference in light of their claim that DDP8/9 processing of AK2 is required for proteasomal degradation?*

These data are steady state protein levels. Even under conditions of DPP9 inhibition, cytosolic AK2 might eventually become degraded (like also IMS-AK2 would). Our data indicate that processing by DPP9 strongly accelerates proteasomal degradation, it is however not an essential prerequisite for degradation as predictions of AK2 stability based on the N-end-rule (ProtParam, ExPASy) show that MAP-cleavage of AK2 already drops the predicted half-life from 30 h to 4.4 h. DPP9-cleavage then further decreases half-life to 1.9 h. Therefore, even in the absence of DPP9-cleavage, AK2 can be expected to be targeted for degradation after some time. Thus, we would expect (and also demonstrate) that inhibiting proteasomal degradation in general exerts a stronger influence on AK2 stability.

4) *The authors nicely show that DDP8/9 can process AK2 to reduce its accumulation in the cytosol. However, the authors do not explain why such cytosolic accumulation can be harmful. It will improve the MS if the authors will indeed test whether such accumulation is harmful.*

We tested in our HEK293-based cell lines whether expression of stable cytosolic AK2 variants impacted cell proliferation. We did this under three different conditions: growth on glucose, growth on galactose, and growth when the carbon source was changed from glucose to galactose (new **Figure S6H**). We thereby find that all cell lines expressing AK2 variants grew similarly to the Mock control. Cells expressing the dominant-negative MIA40^{C53S} variant served as negative control, and these cells are clearly hampered in growth especially during growth on galactose.

We do not think that the accumulation is *per se* harmful, especially not in cells still containing sufficient amounts of AK1 (like HEK293 cells) which appears to be more efficient in catalysing adenylate nucleotide conversion (**S7A,B**). We do think however that there might be conditions, e.g. during immune cell differentiation (some of immune cells contain no or very little AK1) in

which cytosolic accumulation of AK2 might serve a signalling function. We plan on testing this in future studies. At the moment, we are however afraid that such experiments would be outside the scope of this molecular-mechanistic study.

5) *Fig. 4G: the effect of 1G244 in this experiment is not convincing. At Day #2 there is no difference upon addition of the reagent and also in Day #1 the only effect is seen in lane #10. Results that are more convincing should be presented or this part should be omitted.*

We decided to remove this experiment from the manuscript as suggested by the referee. We think that our study convincingly demonstrates the impact of DPP9 on AK2 and mitochondrial biogenesis in HEK293 cells. In a separate study, we will in the future follow up on our investigations in immune cells.

6) *The title is not supported by the actual findings. The authors show neither that the processing by DPP8 is regulated nor that it controls protein import. At the most one can claim that processing competes import. Thus, I suggest to modify the title.*

As suggested by the referee, we modified the title to:

“DPP9-Processing induced Proteasomal degradation competes with mitochondrial protein import”

Minor points:

(a) *Figs. 1B and S1B: Was the immunodetection of HA-AK2 done with antibodies against AK2 or against the HA-tag? If the latter, these should be indicated in the figure. In all IP experiments it should be added to the corresponding legend which percentage of the lysate and eluate were loaded on the gel.*

In **Figure 1B** (and **S1E**), we precipitated MIA40-Strep using Streptactin beads from cells expressing C-terminally Strep tagged MIA40. These cells do only contain endogenous AK2. Thus, in Figure 1B (and S1E), we decorated the immunoblot with an antibody raised against full-length AK2, and we thereby detect endogenous AK2.

In **Figure S1B**, we precipitated AK2-HA using HA-antibody beads from cells expressing C-terminally HA-tagged AK2. These cells do only contain endogenous MIA40. Thus, in Figure S1B, we decorated the immunoblot with an antibody raised against full-length MIA40, and we thereby detect endogenous MIA40.

We now added to the figures and the figure legends the percentage of lysate used as input. We always loaded the complete IP.

(b) *Fig. S1: Using siRNA against AK2-A resulted in a weaker upper band (middle panel). However, this reduction cannot be described as "the upper band disappears". In the right panel the resolution of lane #1 is not good enough to know which, if any, band disappears. Thus, the text should be modified.*

We rephrased the text of the figure legend to **Figure S1A** as follows:

“When we only targeted isoform A the upper band (AK2-A) became weaker (middle panel). When we treated cells with siRNA directed against both isoforms, the AK2 signal was almost completely lost (right panel).”

(c) Fig. 1E: It will be helpful to show a Western of CHCHD4 to estimate the efficiency of the knockdown. In addition, it seems that the oxidation of isoform A is not affected by depletion of CHCHD4. Can the authors explain this observation?

We now provide an immunoblot as new **Figure S1G** that was done on a replica plate of the experiment shown in **Figure 1E**. It shows a decrease in MIA40 levels as well as a decrease in AK2 levels.

Figure 1E: We think that AK2 isoform A is affected by MIA40 depletion. The levels of oxidized isoform A at t=60' are lower than the initial levels of reduced isoform A at t=1' (31% left in Ctrl., and 38% left in siMIA40). Thus, we think that a large part of isoform A is quickly degraded under all conditions and even more so upon MIA40 depletion. We decided to only quantify isoform B, because the resolution in these experiments was often not good enough to distinguish the reduced form of isoform B and the oxidized form of isoform A.

On a different note, we think that there might be isoform-specific differences in the oxidation kinetics because in Figure 1D, the ratio of reduced A and B at 1 min (more A) is very different from the ratio of oxidized A and B at 120 min (more B). It will be interesting to address these differences in future studies.

(d) Page 7: According to the results of Fig. 2 that show rather minor contribution of C40, it is misleading to describe at this stage the influence of C40 as "critical".

We removed the work "critical". C40 appears to exert an influence on the kinetics of import as we demonstrate in **Figure S2**.

(e) Fig. S2D: To evaluate specificity in the binding, the intensities of the bands of CHCHD4 and CPOX (at least in the input lanes) should be similar. Hence another panel should be presented. Does the observation that the input amounts of WT and C40S are similar suggest that interaction with CHCHD4 does not contribute to the overall stability of AK2? The authors should discuss this point.

We now provide another panel with a shorter exposure of the MIA40 signal. The inputs of AK2^{WT} and AK2^{C40S} with respect to the AK2 signal are not similar in **Figure S2D**. Thus, we do think that mutating C40 decreases levels of AK2.

(f) Fig. 3A: lanes 2-3 and 4-5 are labeled the same. Are these two different cell lines for each treatment? This point should be explained in the legend.

We are sorry for the confusion. Two different siRNAs for MIA40 and two different siRNAs for ALR were used in this experiment. This is now clearly indicated in the labelling of the Figure.

(g) Fig. 3D: It seems that WT AK2 is also stabilized upon addition of MG132. On the other side, also in the presence of this proteasome inhibitor the levels of the cysteine mutant is lower than that of the WT protein. Thus, are other proteases (in addition to the proteasome) involved in the removal of the mutant protein? The authors should elaborate on this point.

This is an interesting point, and we agree that other degradation pathways (e.g. autophagy, other proteases) might control the levels of AK2. We did not address this however in the current study as we focused on the effects of DPP9 on AK2 stability.

(h) Fig. 7D, E: The contribution of 1G244 to the total and mitochondrial amounts of NDUFB10-C107S is very similar. Are there in total more mitochondria in 1G244-treated cells?

This is a very interesting point. We tested levels of mitochondrial proteins and compared them to levels of cytosolic proteins in cells treated with 1G244 or left untreated. We thereby could not detect any differences (data not shown).

(i) Page 14, 1st paragraph: Table S1 is mentioned here but this table does not contain information on potential DPP8/9 substrates.

We are sorry for the confusion. We are referring to the accompanying Excel File (Data set EV1, and not to table S1 in the supplemental information.

(j) A list in the Supplementary Information part describing the source of the used antibodies will be helpful.

We included expanded table S1 In the SI and now provide this information.

Thank you for submitting a revised version of your manuscript. It has now been seen by the original referees, whose comments are shown below.

As you will see, they find that criticisms have been sufficiently addressed and recommend the study for publication. However, there are a few editorial issues concerning text and figures that I need you to address, before we can officially accept your manuscript.

Referee #1:

This is a highly exciting manuscript and the authors clarified all issues raised and included many novel experimental data. I have realized that I have been the most critical reviewer, but have to state now that this paper will be a real highlight in the field of mitochondrial protein biogenesis. Congratulations to this milestone paper!

Referee #3:

The authors addressed all my concerns regarding the original submission. In my eyes, the contribution is now suitable for publication in EMBO J.

I am pleased to inform you that your manuscript has been accepted for publication in The EMBO Journal.

USEFUL LINKS FOR COMPLETING THIS FORM

<http://www.antibodypedia.com>
<http://1degreebio.org>
<http://www.equator-network.org/reporting-guidelines/improving-bioscience-research-repor>

<http://grants.nih.gov/grants/olaw/olaw.htm>
<http://www.mrc.ac.uk/Ourresearch/Ethicsresearchguidance/Useofanimals/index.htm>
<http://ClinicalTrials.gov>
<http://www.consort-statement.org>
<http://www.consort-statement.org/checklists/view/32-consort/66-title>

<http://www.equator-network.org/reporting-guidelines/reporting-recommendations-for-tum>

<http://datadryad.org>

<http://figshare.com>

<http://www.ncbi.nlm.nih.gov/gap>

<http://www.ebi.ac.uk/ega>

<http://biomodels.net/>

<http://biomodels.net/miriam/>
<http://jil.biochem.sun.ac.za>
http://oba.od.nih.gov/biosecurity/biosecurity_documents.html
<http://www.selectagents.gov/>

Corresponding Author Name: Jan Riemer
 Journal Submitted to: The EMBO Journal
 Manuscript Number: EMBOJ-2019-103889